# Tackling the Noisy Elephant in the Room: Label Noise-robust Out-of-Distribution Detection via Loss Correction and Low-rank Decomposition

## Abstract

Robust out-of-distribution (OOD) detection is an indispensable component of modern artificial intelligence (AI) systems, especially in safety-critical applications where models must identify inputs from unfamiliar classes not seen during training. While OOD detection has been extensively studied in the machine learning literature—with both post hoc and training-based approaches—its effectiveness under noisy training labels remains underexplored. Recent studies suggest that label noise can significantly degrade OOD performance, yet principled solutions to this issue are lacking. In this work, we demonstrate that directly combining existing label noise-robust methods with OOD detection strategies is insufficient to address this critical challenge. To overcome this, we propose a *robust* OOD detection framework designed to *cleanse* feature embeddings, thereby mitigating the adverse effects of noisy labels on OOD performance. Towards this, we introduce an end-to-end training strategy that integrates loss correction methods from the noisy-label learning literature with low-rank and sparse decomposition techniques from signal processing. Building on this strategy, we derive a novel metric that quantifies the "OOD-ness" content within training data, which in turn leads to a label noise-robust OOD detection scoring technique. Extensive experiments on both synthetic and real-world datasets demonstrate that our method significantly outperforms the state-of-the-art OOD detection techniques, particularly under severe noisy label settings.

## 1 Introduction

Artificial intelligence (AI) models have achieved remarkable performance across myrid of domains including computer vision and natural language processing. Yet, a persistent challenge arises in real-world deployment: these models often fail to recognize inputs from unfamiliar data distributions, leading to overly confident and potentially misleading predictions (Goodfellow et al., 2014). This limitation underscores the importance of out-of-distribution (OOD) detection for building trustworthy AI systems, particularly in high-stakes domains such as autonomous driving (Geiger et al., 2012) and medical diagnostics (Schlegl et al., 2017). The goal of OOD detection is not only to provide accurate prediction on seen data distributions but also to flag inputs from novel or unobserved distributions (Hendrycks & Gimpel, 2016).

OOD detection has been an active topic of research in the field of AI for many decades; a recent survey can be found in (Yang et al., 2024). A key focus in this field is detecting semantic shifts—scenarios where new, previously unseen classes appear in the test data, resulting in a mismatch between the label spaces of in-distribution (ID) and OOD samples. A wide range of methods have been proposed for OOD detection, including softmax/logit-based post-hoc techniques (Hendrycks & Gimpel, 2016; Liang et al., 2018; Hendrycks et al., 2022; Sun & Li, 2022; Sun et al., 2021; Dong et al., 2022) and feature distance-based strategies (Lee et al., 2018; Sun et al., 2022; Ming et al., 2023; Sehwag et al., 2021; Ghosal et al., 2024). As softmax/logit-based approaches are often susceptible to overconfidence, recent research has increasingly focused on distance-based methods—such as Mahalanobis distance (Lee et al., 2018) and $k$-nearest neighbors ($k$NN)(Sun et al.,

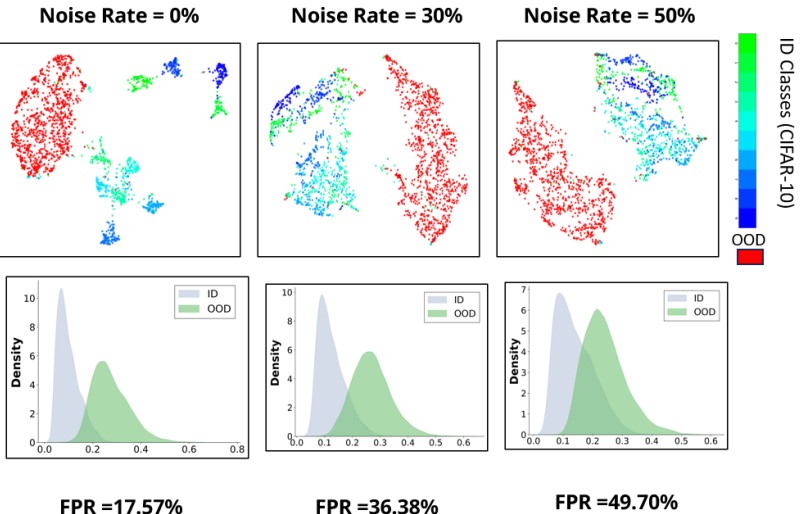

Figure 1: The upper row presents the UMAP representations of the latent feature vectors $h(x)$ learned using the cross-entropy loss–based training on the noisily labeled dataset $\{x_n, \widehat{y}_n\}$ for various synthetic noise rates. The clusters are more distorted for the training data, losing the ID-ness characteristics, resulting in degraded performance in OOD detection during test time. The lower row presents the corresponding $k$NN score histogram for the ID and OOD datasets. The false positive ratio (FPR) for OOD detection using the $k$NN score is also reported.

2022)—due to their empirical success. These methods rely on the observation that feature representations extracted from OOD samples typically lie farther from the ID feature space. Since deep neural networks naturally encode semantic similarity in their embeddings—forming well-structured clusters for similar samples—distance-based approaches leverage this structure to better separate ID and OOD samples. Nonetheless, most of these methods are developed under the assumption that models are trained on *clean, correctly* labeled data. However, in practice, training datasets often contain noisy labels, stemming from the scarcity of expert annotators and the high cost of accurate label acquisition (Buhrmester et al., 2016). Recent empirical studies have brought serious attention to this issue, revealing that the presence of label noise can significantly degrade the performance of state-of-the-art OOD detection methods (Humblot-Renaux et al., 2024). This highlights a critical gap in current research and underscores the need to develop robust OOD detection frameworks that remain reliable under real-world label noise.

The effect of label noise on training deep learning models has been extensively studied in recent years; see the survey (Song et al., 2022). It is now well-established that training deep neural network (DNN) models with noisy labels can severely degrade classification performance, leading to poor generalization and overfitting (Arpit et al., 2017; Zhang et al., 2016). To address this, a variety of label noise-robust methods have been proposed, including loss correction strategies such as probabilistic modeling techniques (Liu & Tao, 2016; Patrini et al., 2017; Li et al., 2021b; Xia et al., 2020; Yang et al., 2021; Cheng et al., 2020), robust loss function designs (Zhang & Sabuncu, 2018a; Lyu & Tsang, 2019; Wang et al., 2019a), and in-built sample selection strategies (Jiang et al., 2018; Yu et al., 2019; Nguyen et al., 2019; Han et al., 2018a; Li et al., 2020b). However, their effectiveness in OOD detection when the training labels are noisy remains largely unexplored. The key challenge lies in the misalignment of objectives: while label noise methods aim to correct the prediction probabilities within the training distribution, OOD detection requires learning discriminative feature representations to detect the samples that does not belong to the training distribution. Hence, most existing label-noise approaches exhibit poor OOD detection performance when applied directly, as we will demonstrate in detail in subsequent sections.

**Our Contributions.** In this work, we investigate the critical challenge of robust OOD detection in the presence of noisy labels in the training set. Unlike existing studies that focus solely on the empirical limitations of current OOD detection methods (Humblot-Renaux et al., 2024), we identify a key

gap: the label noise-robust methods improves generalization under noisy supervision for classification settings, yet they are largely ineffective when directly applied for OOD detection. To address this limitation, we have the following key contributions in this work:

- *Robust OOD detection framework.* We propose a novel distance-based OOD detection framework that effectively cleanses feature embeddings, enhancing robustness under noisy training labels. Our framework is accompanied by an easy-to-implement, end-to-end learning criterion, leveraging the loss correction techniques and low-rank and sparse decomposition strategies.

- *Novel OOD-detection scoring.* The framework allows us to quantify the "OOD-ness" content of each training sample, producing a score that reflects its difficulty in training the OOD detector. Leveraging this score, we introduce an enhanced distance-based scoring function that is more robust compared to existing metrics.

- *Promising empirical evidence for OOD detection.* We conduct extensive experiments using synthetic noise settings, several real-world noisy datasets, e.g., CIFAR-10N, CIFAR-100N (Wei et al., 2022), and Animal10N (Song et al., 2019), and evaluate our approach across diverse OOD datasets. Additionally, we perform detailed ablation studies on key hyperparameters to demonstrate the robustness of our approach.

To the best of our knowledge, this work is the first to offer a principled solution to the problem, achieving substantial improvements over state-of-the-art OOD detection methods in the presence of label noise.

## 2 PROBLEM STATEMENT

Consider an input feature space $\mathcal{X} \subset \mathbb{R}^D$, where $D$ denotes the dimensionality of the input features. Let the label space be defined as $\mathcal{Y} = \{1, \ldots, K\}$, corresponding to $K$ distinct classes for the ID data. We define the training dataset $\mathcal{D}$ as $\mathcal{D} = \{(\boldsymbol{x}_n, y_n)\}_{n=1}^N, \quad \boldsymbol{x}_n \in \mathcal{X}, \quad y_n \in \mathcal{Y}$, where $\boldsymbol{x}_n$ is the feature vector of the $n$-th training example, $y_n$ is its associated *ground-truth* class label, and $N$ denotes the total number of training samples. Each pair $(\boldsymbol{x}_n, y_n)$ is assumed to be drawn independently and identically distributed (i.i.d.) from an underlying joint distribution $\mathcal{P}_{\mathcal{X}\mathcal{Y}}$. Let $\boldsymbol{h} : \mathbb{R}^D \to \mathbb{R}^L$ denote a DNN that maps each input $\boldsymbol{x}_n$ to an $L$-dimensional *latent feature* representation $\boldsymbol{h}(\boldsymbol{x}_n)$. For the task of multi-class classification, we employ a projection head $\boldsymbol{c} : \mathbb{R}^L \to \mathbb{R}^K$ to produce pre-softmax logits. Thus, the overall label prediction function is given by $\boldsymbol{f}(\boldsymbol{x}_n) = \boldsymbol{\sigma}(\boldsymbol{c}(\boldsymbol{h}(\boldsymbol{x}_n)))$, where $\boldsymbol{\sigma}$ denotes the softmax function that output the class probabilities. We often learn the parameters of these functions by training via cross-entropy (CE) minimization as follows:

$$\text{minimize}_{\boldsymbol{\theta}} \ \mathcal{L}_{\text{CE}}(\boldsymbol{\theta}; \{\boldsymbol{x}_n, y_n\}) = -\sum_{n=1}^N \sum_{k=1}^K \mathbb{I}[y_n = k] \log(\boldsymbol{f}(\boldsymbol{x}_n)))), \tag{1}$$

where $\boldsymbol{\theta}$ denotes the DNN parameters of both $\boldsymbol{h}$ and $\boldsymbol{c}$ functions.

**OOD Detection.** AI systems are generally learned under the closed-world assumption, where it is presumed that test samples are drawn from the same distribution as the training data. However, this assumption often fails in practical scenarios, where models inevitably encounter samples that lie outside the training distribution. These unfamiliar samples are known as OOD inputs (Hendrycks & Gimpel, 2016). In classification tasks, such distributional shift may manifest as a semantic shift, wherein some test instances originate from an *unknown* label space $\mathcal{Y}^o$, disjoint from the known label space, i.e., $\mathcal{Y} \cap \mathcal{Y}^o = \emptyset$. The objective of OOD detection is to identify whether a given test input belongs to the in-distribution (ID) or not, thereby preventing the model from making confident predictions on OOD inputs. Thus, OOD detection can be considered as a binary classification task that distinguishes ID samples from OOD ones. This can be formalized by a detection function:

$$\boldsymbol{g}_\tau(\boldsymbol{x}) = \begin{cases} \mathsf{ID} & \text{if } s(\boldsymbol{x}) \geq \tau, \\ \mathsf{OOD} & \text{if } s(\boldsymbol{x}) < \tau, \end{cases} \tag{2}$$

where $s(\boldsymbol{x})$ is a scoring function that quantifies the likelihood of $\boldsymbol{x}$ belonging to the ID distribution, and $\tau$ is a predefined threshold.

Typically, scoring function $s(\boldsymbol{x})$ is derived from the trained parameters of the underlying DNN model. Several scoring functions have been proposed in the literature. Early OOD detection methods directly used the softmax outputs $\boldsymbol{f}(\boldsymbol{x})$ to score "OOD-ness" (Hendrycks & Gimpel, 2016; Liang et al., 2018), but they suffered from overconfidence issues, reducing the desired ID-OOD separability. Further, pre-softmax activations-based approaches (e.g., by using the logits $\boldsymbol{c}(\boldsymbol{h}(\boldsymbol{x}))$) were introduced (Hendrycks et al., 2022; Sun & Li, 2022; Sun et al., 2021; Dong et al., 2022), though they remained sensitive to architecture and still faced overconfidence issues. Recently, distance-based methods such as those based on Mahalanobis (Lee et al., 2018) and $k$-nearest neighbor ($k$NN) (Sun et al., 2022; Ming et al., 2023; Sehwag et al., 2021; Ghosal et al., 2024) have gained traction by leveraging the clusterability of latent feature representations $\boldsymbol{h}(\boldsymbol{x})$. In essence, the success of OOD detection lies in the careful design and learning of the scoring function $s(\boldsymbol{x})$ that can ensure the ID-OOD separability during test time.

**Learning under Label Noise.** Most studies in the domain of OOD detection assume that the DNN classifier $\boldsymbol{f}$ and the scoring function $s$ are learned using ground-truth labels $y_n$. However, the lack of access to reliable ground-truth annotations is a significant challenge for robust OOD detection–see an example in Fig. 1 where the clusterability of the latent representations $\boldsymbol{h}(\boldsymbol{x})$ is severely compromised under label noise, leading to significant degradation in ID-OOD separability for the $k$NN score function. In scenarios where ground-truth labels $y_n$ are difficult to obtain, we often rely on their noisy counterparts, denoted by $\widehat{y}_n \in \{1, \ldots, K\}$, associated with each data item $\boldsymbol{x}_n$. In noisy label settings, for many data items, the observed label does not match the true label, i.e., $\widehat{y}_n \neq y_n$.

> The goal of *label noise-robust OOD detection* is two-fold: *(i)* accurately classify ID samples through a well-generalized predictor $\boldsymbol{f}$, and *(ii)* reliably detect OOD instances using a robust decision function $\boldsymbol{g}_\tau$, despite learning them using the noisily labeled dataset $\widehat{\mathcal{D}} = \{(\boldsymbol{x}_n, \widehat{y}_n)\}_{n=1}^N$, $\boldsymbol{x}_n \in \mathcal{X}$, $\widehat{y}_n \in \mathcal{Y}$.

## 3 PROPOSED APPROACH

In this section, we present our label noise-robust OOD detection framework. Our strategy is based on *cleansing* the noise-corrupted latent feature space using an end-to-end training strategy, thereby making it robust for OOD detection at test time. Towards this goal, our framework encompasses three main components: *i)* loss correction module *ii)* low rank and sparse decomposition of latent feature matrix *iii)* robust OOD detector using distance-based metrics, e.g., $k$NN.

**Loss Correction.** As demonstrated in Fig. 1, training directly with noisy labels (e.g., by using the CE minimization as in equation 1 where the unobserved ground-truth labels $y_n$ are replaced by the observed noisy labels $\widehat{y}_n$) leads to a corrupted latent feature space. To address this, we first integrate a loss correction module to reduce the effect of label noise in learned features. Loss correction strategies have attracted considerable attention in noisy label learning literature. Among these, probabilistic noise modeling via the so-called *transition matrices* (Patrini et al., 2017; Tanno et al., 2019; Li et al., 2021b; Ibrahim et al., 2023) and robust loss function-based approaches (Zhang & Sabuncu, 2018a; Lyu & Tsang, 2019; Wang et al., 2019a) are particularly well-received, owing to their strong theoretical foundations and robust empirical performance in classification tasks.

In general, loss correction strategies design a modified cross-entropy loss to train the classifier $\boldsymbol{f}$ on the noisy dataset $\{\boldsymbol{x}_n, \widehat{y}_n\}$, while aiming to predict the ground-truth labels, i.e.,

$$\underset{\boldsymbol{\theta}, \boldsymbol{\eta}}{\text{minimize}} \; \mathcal{L}_{\text{CE}}^{\text{mod}}(\boldsymbol{\theta}, \boldsymbol{\eta}; \{\boldsymbol{x}_n, \widehat{y}_n\}) \tag{3}$$

where $\boldsymbol{\eta}$ typically refers to additional model parameters according to specific loss designs. For instance, in the case of transition matrix-based approaches (Patrini et al., 2017; Tanno et al., 2019; Li et al., 2021b; Ibrahim et al., 2023), $\boldsymbol{\eta}$ refers to the noise transition probabilities that learns the probabilistic label confusion *confusion matrix* $\boldsymbol{T}$, whose $(k, k')$-th entry models the label corruption process via $[\boldsymbol{T}]_{k,k'} = \Pr(\widehat{y}_n = k | y_n = k')$. Intuitively, $\boldsymbol{T}$ describes how clean labels $y_n$ are stochastically flipped into noisy labels $\widehat{y}_n$. The modified cross-entropy then incorporates $\boldsymbol{T}$ by mapping the classifier's predicted clean-label distribution into the corresponding noisy-label distribution, so that training can directly account for and correct the effect of label noise. In sample selection approaches (Jiang et al., 2018; Yu et al., 2019; Nguyen et al., 2019; Han et al., 2018a; Li et al., 2020b) , $\boldsymbol{\eta}$ instead represents sample-weighting terms that regulate the contributions of clean and noisy sample–label

pairs. In contrast, robust loss function-based methods, e.g., (Zhang & Sabuncu, 2018a; Wang et al., 2019a), often do not introduce additional parameters—they directly design loss functions that are inherently less sensitive to incorrect labels. Popular approaches such as symmetric cross-entropy (SCE) (Wang et al., 2019a) and generalized cross-entropy (GCE) (Zhang & Sabuncu, 2018a) can be viewed as hybrids of CE loss and mean absolute error (MAE) loss, thereby combining the favorable convergence properties of CE with the robustness of MAE against outliers. For instance, the SCE loss function is defined as follows:

$$\ell_{\mathsf{SCE}}(\boldsymbol{g}, \boldsymbol{y}) = -\alpha \sum_{k=1}^{K} [\boldsymbol{y}]_k \log[\boldsymbol{g}]_k - \beta \sum_{k=1}^{K} [\boldsymbol{g}]_k \log[\boldsymbol{y}_\epsilon]_k,$$

where the parameters $\alpha, \beta > 0$ are certain scalars, $\boldsymbol{g} \in \mathbb{R}^K$ is a probability vector such that $\boldsymbol{g} \geq \boldsymbol{0}, \boldsymbol{1}^\top \boldsymbol{g} = 1, \boldsymbol{y} \in \{0,1\}^K$ is a one-hot label, and $\boldsymbol{y}_\epsilon$ takes care of the $\log 0$ cases of the one-hot label $\boldsymbol{y}$ using a small constant $\epsilon$. Note that the first term in the R.H.S is the scaled CE loss, whereas the second term in the R.H.S reduces to the MAE loss when $\epsilon = -2$ (Wang et al., 2019a).

Nonetheless, these loss correction strategies primarily operate by modifying the softmax prediction outputs of the ID samples rather than directly *correcting* their feature embeddings. However, feature embeddings are often more critical for OOD detection, particularly for the distance-based OOD metrics such as $k$-nearest neighbor (Sun et al., 2022) and Mahalanobis (Lee et al., 2018). This misalignment of objectives results in suboptimal performance in mitigating the effect of label noise in OOD detection.

**Low-rank plus Sparse Decomposition.** To overcome the limitation of loss correction modules in handling feature correction, we introduce the next key component of our framework. A critical observation underlying its design is that, in the absence of label noise, latent feature vectors naturally exhibit certain clustering patterns, reflecting their low-rank structure due to their class-specific organization—see the first UMAP plot in Fig. 1. This intrinsic structural tendency can be explicitly leveraged in the training phase to encourage low-rank properties in the feature representations. To this end, we adopt a low-rank and sparse decomposition strategy, drawing inspiration from classical signal processing techniques (Candès et al., 2011; Zhang et al., 2011).

Consider the latent feature representation $\boldsymbol{h}(\boldsymbol{x})$ of the input image $\boldsymbol{x}$ (e.g., the penultimate layer encoding of the DNN model). Let us represent the latent feature matrix $\boldsymbol{H}$ as follows: $\boldsymbol{H} = \left[ \boldsymbol{h}(\boldsymbol{x}_1), \ldots, \boldsymbol{h}(\boldsymbol{x}_N) \right] \in \mathbb{R}^{D \times N}$, where $D$ is the feature dimension and $N$ is the number of training samples. Our goal is to promote a low-rank structure in this feature matrix (as typically observed when labels are clean), while assuming that samples that deviate strongly from this structure (e.g., difficult or confusing examples that much deviate from the ID-ness characteristics) are relatively few. Hence, in order to exploit the low-rankness of the latent matrix $\boldsymbol{H}$ along with a sparse structure, we assume that $\boldsymbol{H} \approx \boldsymbol{L} + \boldsymbol{S}$, where $\boldsymbol{L} \in \mathbb{R}^{D \times N}$ is the low-rank component and $\boldsymbol{S} \in \mathbb{R}^{D \times N}$ is a column sparse matrix, i.e., most columns of $\boldsymbol{S}$ have zero $\ell_2$ norm. That means, the low-rank term captures the underlying class structure information, whereas the sparse term can handle the outlier data items that do not strictly conform to the low-rank assumption.

Learning $\boldsymbol{L}$ and $\boldsymbol{S}$ from the observed matrix $\boldsymbol{H}$ generally involves solving an optimization problem of the form (Candes et al., 2009; Wang et al., 2008):

$$\min_{\boldsymbol{L}, \boldsymbol{S}} \ \|\boldsymbol{L}\|_* + \lambda \|\boldsymbol{S}\|_{2,1} \quad \text{s.t.} \quad \boldsymbol{H} = \boldsymbol{L} + \boldsymbol{S},$$

where $\|\boldsymbol{L}\|_*$ denotes the nuclear norm of $\boldsymbol{L}$ to promote the low-rankness and $\|\boldsymbol{S}\|_{2,1}$ denotes the matrix mixed norm that promotes column sparsity in $\boldsymbol{S}$. Here, $\lambda > 0$ is a regularization parameter that balances the contributions of the low-rank and sparse terms. Directly minimizing the nuclear norm can be computationally expensive, as it typically requires repeated singular value decompositions. To avoid this cost during training, we adopt an efficient power-iteration (PI)–based low-rank approximation strategy (Rokhlin et al., 2010; Gu, 2015). Specifically, The method estimates the top-$K$ left singular vectors of the latent representation $\boldsymbol{H}$ by iteratively refining a randomly initialized orthonormal basis $\boldsymbol{Q} \in \mathbb{R}^{D \times K}$ through alternating projections of $\boldsymbol{H}$ and orthonormalization via QR decomposition. Here, The rank $K$ can be selected according to the number of classes (or based on the number of coarse-grained classes in the case of datasets with very large label space). Using the learned $\boldsymbol{Q}$, we decompose the feature matrix as

$$\boldsymbol{H}_{\text{ID}} = [\boldsymbol{h}_{\text{ID}}(\boldsymbol{x}_1), \ldots, \boldsymbol{h}_{\text{ID}}(\boldsymbol{x}_N)] = (\boldsymbol{Q}\boldsymbol{Q}^\top)\boldsymbol{H}, \ \ \boldsymbol{H}_{\text{OOD}} = [\boldsymbol{h}_{\text{OOD}}(\boldsymbol{x}_1), \ldots, \boldsymbol{h}_{\text{OOD}}(\boldsymbol{x}_N)] = \boldsymbol{H} - \boldsymbol{H}_{\text{ID}},$$

where $\boldsymbol{H}_{\text{ID}} \in \mathbb{R}^{D \times N}$ represents the ID component and $\boldsymbol{H}_{\text{OOD}} \in \mathbb{R}^{D \times N}$ contains residual features that may potentially carry non-ID information (or OOD-like components). Further, to enforce the column sparsity in the matrix $\boldsymbol{H}_{\text{OOD}}$, we employ the following regularization term:

$$\mathcal{L}_{\text{sparse}} = \|\boldsymbol{H}_{\text{OOD}}\|_{2,1} = \sum_{j=1}^{N} \sqrt{\sum_{i=1}^{D} (\boldsymbol{H}_{\text{OOD}})_{ij}^2}. \tag{4}$$

Note that the outlier content captured in $\boldsymbol{H}_{\text{OOD}}$ matrix can include both spurious features learned due to the influence of noisily labeled training examples as well as the feature components within the correctly labeled training samples that deviate from ID characteristics. We do not attempt to distinguish between these components explicitly, instead $\boldsymbol{H}_{\text{OOD}}$ is designed to capture anything that is non-ID, while the low-rank component $\boldsymbol{H}_{\text{OOD}}$ focuses on learning the cleanest possible version of the ID feature space.

Finally, the proposed method is trained by minimizing a joint objective that combines the modified cross-entropy loss as explained in equation 3 with the regularizer in equation 4:

$$\mathcal{L}_{\mathcal{F}} = \mathcal{L}_{\text{CE}}^{\text{mod}} + \lambda \, \mathcal{L}_{\text{sparse}},$$

where $\lambda > 0$ is a regularization hyperparameter that controls the strength of the column-sparsity term. The detailed algorithm is presented in the supplementary section.

**Remark 1** *The added regularization as described in equation 4 is implemented via the PI–based low-rank approximation of the batch-wise feature matrix $\boldsymbol{H}$ (see the detailed description in Algorithm 1). Since the PI method uses simple matrix–vector multiplications instead of a full SVD, the additional computational complexity is approximately $\mathcal{O}\big(T(DBK) + DK^2\big)$, where $B$ is the batch size, $K$ is the number of classes (or the chosen rank parameter), $D$ is the feature embedding dimension, and $T$ is the number of PI iterations, which is often a smaller number in practice. This overhead is moderate and substantially cheaper than exact nuclear-norm optimization for low-rank decompositions.*

**Robust OOD Detection with Refined Feature Representations.** To detect OOD samples at test time, we propose a feature distance–based score with a robust selection strategy for reference embeddings. Specifically, for each training sample $\boldsymbol{x}_n$, we assign an OOD-ness score defined as $o(\boldsymbol{x}_n) = \|\boldsymbol{h}(\boldsymbol{x}_n) - \boldsymbol{h}_{\text{ID}}(\boldsymbol{x}_n)\|_2$, where larger scores indicate greater deviation from ID characteristics (e.g., noisy or ambiguous samples), while smaller scores correspond to clean, representative ID examples—also see Fig. 3a in Sec. 4. Based on this score, we retain only the most challenging examples by selecting

$$\mathcal{S} = \{n \in [N] \mid o(\boldsymbol{x}_n) \geq \alpha_p\}, \qquad \alpha_p = \text{Quantile}_{(1-p)}\big(\{o(\boldsymbol{x}_n)\}_{n=1}^N\big), \tag{5}$$

so that the top $(1-p)\%$ of samples with the highest OOD-ness scores are used to construct a robust detector, where $p \in [0, 1)$. For each selected sample $n \in \mathcal{S}$, we compute the $\ell_2$-normalized feature vectors $\boldsymbol{u}_{\text{ID}}(\boldsymbol{x}_n) = \boldsymbol{h}_{\text{ID}}(\boldsymbol{x}_n)/\|\boldsymbol{h}_{\text{ID}}(\boldsymbol{x}_n)\|_2$. Similarly, the test sample $\boldsymbol{x}^*$ is mapped to its normalized feature $\boldsymbol{u}(\boldsymbol{x}^*) = \boldsymbol{h}(\boldsymbol{x}^*)/\|\boldsymbol{h}(\boldsymbol{x}^*)\|_2$, whose distance to the stored ID embeddings $\{\boldsymbol{u}_{\text{ID}}(\boldsymbol{x}_n)\}_{n \in \mathcal{S}}$ is then evaluated. For example, using $k$NN distance, we have the score defined as $s_{\text{kNN}}(\boldsymbol{x}^*) = -\big\|\boldsymbol{u}(\boldsymbol{x}^*) - \boldsymbol{u}_{\text{ID}}^{(k)}\big\|_2$, where $\boldsymbol{u}_{\text{ID}}^{(k)}$ denotes the $k$-th nearest embedding in $\mathcal{S}$. A similar modified metric can be derived using Mahalanobis distance as well. A decision threshold $\tau$ of the detection function $g_\tau$ is chosen based on a validation set to ensure a high ID recall (e.g., 95%).

## 4 EXPERIMENTS

In this section, we evaluate the effectiveness of our label noise-robust OOD detection framework, which is named as **N**oise-robust **O**ut-**O**f-**D**istribution **Le**arning (NOODLE)

**Datasets.** For synthetic label noise settings, we consider CIFAR-10 (Krizhevsky, 2009) as ID dataset. For synthetic label noise generation, we adopt class-independent symmetric noise, where every true label has the same probability of being corrupted, and when corrupted, it is flipped uniformly at random to any of the other $K - 1$ classes, regardless of the original class. We vary the

Table 1: OOD detection performance on CIFAR10 with synthetic label noise across different OOD datasets; The top two performing algorithms (in terms of average FPR95) are highlighted in bold.

| Method | SVHN | | FashionMNIST | | LSUN | | iSUN | | DTD/Texture | | Places365 | | Average | |
|---|---|---|---|---|---|---|---|---|---|---|---|---|---|---|
| | FPR95↓ | AUROC↑ | FPR95 | AUROC | FPR95 | AUROC | FPR95 | AUROC | FPR95 | AUROC | FPR95 | AUROC | FPR95 | AUROC |
| *Noise rate = 10%* | | | | | | | | | | | | | | |
| *OOD baselines* | | | | | | | | | | | | | | |
| KNN | 34.23 | 93.92 | 38.51 | 93.57 | 17.71 | 96.66 | 21.72 | 95.93 | 28.58 | 94.04 | 62.26 | 84.62 | 33.84 | 93.12 |
| MSP | 73.54 | 84.82 | 54.92 | 88.64 | 32.84 | 94.49 | 55.18 | 89.70 | 77.02 | 75.86 | 69.01 | 80.33 | 60.42 | 85.64 |
| ODIN | 87.82 | 60.19 | 64.22 | 79.77 | 24.42 | 94.56 | 32.99 | 91.49 | 80.30 | 56.06 | 78.84 | 66.87 | 61.43 | 74.82 |
| Energy | 80.31 | 77.93 | 57.89 | 84.84 | 17.74 | 96.65 | 54.63 | 88.38 | 82.43 | 62.42 | 78.29 | 72.39 | 60.37 | 79.12 |
| ReAct | 96.77 | 53.22 | 63.00 | 87.87 | 49.38 | 90.45 | 72.98 | 81.08 | 92.89 | 44.08 | 75.21 | 75.29 | 75.04 | 72.00 |
| Mahalanobis | 31.71 | 91.10 | 72.56 | 74.47 | 28.29 | 93.87 | 52.77 | 81.81 | 49.11 | 80.53 | 94.74 | 44.55 | 54.86 | 77.72 |
| CIDER | 30.04 | 95.15 | 20.78 | 96.62 | 35.14 | 93.34 | 66.61 | 84.09 | 46.67 | 92.82 | 62.94 | 88.58 | 43.69 | 91.77 |
| SSD+ | 52.65 | 87.82 | 80.75 | 85.52 | 90.05 | 77.82 | 65.24 | 84.64 | 21.60 | 95.72 | 16.22 | 96.88 | 54.42 | 88.07 |
| SNN | 49.71 | 91.73 | 29.72 | 95.06 | 20.96 | 96.25 | 25.36 | 95.23 | 34.26 | 92.53 | 56.29 | 86.46 | 36.05 | 92.88 |
| *Label noise-robust baselines* | | | | | | | | | | | | | | |
| SCE | 5.97 | 98.96 | 12.60 | 97.80 | 2.93 | 99.44 | 15.62 | 97.01 | 27.16 | 94.28 | 59.64 | 85.79 | 20.66 | 95.55 |
| GCE | 7.81 | 98.50 | 16.20 | 97.33 | 5.88 | 98.96 | 12.03 | 97.92 | 31.63 | 93.56 | 47.83 | 89.35 | 20.23 | 95.94 |
| Co-teaching | 40.12 | 90.87 | 99.29 | 59.93 | 75.83 | 76.34 | 96.95 | 54.21 | 49.45 | 82.48 | 93.19 | 48.57 | 75.81 | 68.73 |
| DivideMix | 62.65 | 87.37 | 68.10 | 81.99 | 49.01 | 91.24 | 42.84 | 91.52 | 37.48 | 92.79 | 77.16 | 75.03 | 56.21 | 86.66 |
| CM | 6.14 | 98.90 | 15.80 | 97.05 | 6.03 | 98.91 | 10.40 | 98.12 | 19.08 | 96.47 | 53.54 | 87.17 | 18.50 | 96.10 |
| VolMinNet | 2.64 | 99.47 | 5.18 | 98.94 | 5.00 | 98.98 | 9.26 | 98.24 | 21.60 | 95.86 | 56.16 | 87.43 | **16.64** | 96.49 |
| NOODLE | 3.51 | 99.28 | 5.03 | 98.97 | 3.31 | 99.33 | 3.05 | 99.22 | 16.61 | 96.78 | 48.02 | 89.62 | **13.26** | 97.20 |
| *Noise rate = 30%* | | | | | | | | | | | | | | |
| *OOD baselines* | | | | | | | | | | | | | | |
| KNN | 23.80 | 95.80 | 36.15 | 93.43 | 27.04 | 94.61 | 22.03 | 95.86 | 39.50 | 90.40 | 69.76 | 83.53 | 36.38 | 92.27 |
| MSP | 76.88 | 80.34 | 56.16 | 87.14 | 29.90 | 93.92 | 58.82 | 88.30 | 79.04 | 71.54 | 74.98 | 76.61 | 62.63 | 82.97 |
| ODIN | 83.79 | 61.60 | 50.70 | 83.57 | 22.26 | 94.83 | 35.64 | 89.98 | 79.68 | 54.97 | 83.19 | 60.62 | 59.21 | 74.26 |
| Energy | 76.67 | 76.82 | 51.45 | 86.81 | 19.38 | 95.65 | 66.77 | 84.97 | 80.89 | 62.95 | 77.32 | 72.32 | 62.08 | 79.92 |
| ReAct | 88.89 | 67.41 | 62.61 | 85.50 | 21.63 | 95.34 | 90.19 | 65.23 | 91.33 | 51.51 | 82.51 | 68.95 | 72.86 | 72.33 |
| Mahalanobis | 37.86 | 90.25 | 50.42 | 85.21 | 26.98 | 93.29 | 60.92 | 79.60 | 52.75 | 77.45 | 95.94 | 40.86 | 54.15 | 77.78 |
| CIDER | 90.69 | 65.27 | 60.75 | 79.74 | 96.69 | 48.38 | 90.98 | 66.91 | 59.75 | 76.81 | 70.11 | 79.49 | 78.16 | 69.43 |
| SSD+ | 76.34 | 78.07 | 61.81 | 88.45 | 90.07 | 77.66 | 84.61 | 79.29 | 28.94 | 94.78 | 50.55 | 87.94 | 65.39 | 84.37 |
| SNN | 23.37 | 95.72 | 34.55 | 94.11 | 25.91 | 94.90 | 34.38 | 92.36 | 42.27 | 89.39 | 65.06 | 84.19 | 37.59 | 91.78 |
| *Label noise-robust baselines* | | | | | | | | | | | | | | |
| SCE | 19.48 | 96.45 | 25.84 | 95.37 | 16.58 | 96.57 | 61.71 | 87.42 | 35.50 | 91.76 | 74.66 | 79.27 | 38.96 | 91.14 |
| GCE | 58.38 | 91.40 | 20.37 | 96.64 | 11.32 | 97.95 | 12.38 | 97.73 | 30.53 | 94.08 | 51.59 | 88.18 | 30.76 | 94.33 |
| Co-teaching | 50.10 | 83.16 | 99.99 | 20.37 | 96.73 | 64.14 | 97.86 | 41.70 | 53.71 | 79.45 | 93.84 | 48.63 | 82.04 | 56.24 |
| DivideMix | 58.39 | 90.07 | 31.17 | 94.93 | 27.86 | 95.59 | 16.38 | 96.93 | 36.28 | 92.76 | 59.28 | 84.34 | 38.22 | 92.44 |
| CM | 22.04 | 96.76 | 8.79 | 98.04 | 10.17 | 98.12 | 23.30 | 95.64 | 23.71 | 94.99 | 55.42 | 86.90 | **23.90** | 95.08 |
| VolMinNet | 4.99 | 99.04 | 14.01 | 97.09 | 9.48 | 98.33 | 51.23 | 89.68 | 27.84 | 93.44 | 59.25 | 85.42 | 27.80 | 93.84 |
| NOODLE | 1.84 | 99.60 | 19.66 | 96.36 | 7.28 | 95.53 | 10.76 | 97.89 | 20.67 | 95.85 | 57.50 | 85.87 | **19.62** | 95.68 |
| *Noise rate = 50%* | | | | | | | | | | | | | | |
| *OOD baselines* | | | | | | | | | | | | | | |
| KNN | 65.53 | 85.64 | 37.84 | 93.71 | 30.61 | 93.38 | 45.41 | 89.21 | 43.81 | 89.06 | 74.98 | 79.58 | 49.70 | 88.43 |
| MSP | 96.92 | 53.68 | 80.68 | 77.70 | 47.78 | 89.67 | 67.84 | 83.23 | 82.50 | 68.45 | 81.12 | 73.63 | 76.14 | 74.40 |
| ODIN | 94.94 | 44.15 | 71.46 | 80.28 | 34.04 | 91.82 | 47.51 | 88.11 | 79.61 | 60.80 | 82.87 | 66.17 | 68.40 | 71.89 |
| Energy | 97.93 | 46.79 | 83.41 | 76.67 | 39.77 | 90.87 | 67.60 | 81.00 | 85.43 | 60.47 | 82.28 | 69.91 | 76.07 | 70.95 |
| ReAct | 99.19 | 24.75 | 90.11 | 64.20 | 50.47 | 85.45 | 78.12 | 67.79 | 93.79 | 39.56 | 86.40 | 62.25 | 83.02 | 57.33 |
| Mahalanobis | 55.77 | 83.12 | 59.93 | 85.98 | 31.23 | 93.54 | 45.17 | 88.63 | 48.90 | 81.70 | 93.28 | 51.47 | 55.71 | 80.74 |
| CIDER | 99.02 | 50.38 | 97.54 | 52.78 | 91.99 | 48.96 | 98.67 | 45.37 | 99.16 | 50.86 | 92.60 | 59.47 | 96.50 | 51.30 |
| SSD+ | 83.54 | 77.49 | 76.76 | 76.56 | 83.89 | 76.28 | 86.10 | 73.24 | 62.31 | 81.41 | 42.27 | 88.67 | 72.48 | 78.94 |
| SNN | 71.41 | 83.96 | 68.22 | 87.66 | 53.56 | 89.27 | 63.49 | 80.99 | 56.95 | 85.48 | 82.28 | 77.12 | 65.99 | 84.08 |
| *Label noise-robust baselines* | | | | | | | | | | | | | | |
| SCE | 14.10 | 97.40 | 42.30 | 90.96 | 25.18 | 94.09 | 67.17 | 80.81 | 51.51 | 84.65 | 70.67 | 77.75 | 45.15 | 87.61 |
| GCE | 19.19 | 96.43 | 29.10 | 95.06 | 22.98 | 95.40 | 53.92 | 86.31 | 48.35 | 87.54 | 65.83 | 83.58 | 39.89 | 90.72 |
| Co-teaching | 57.05 | 76.43 | 99.97 | 28.21 | 99.23 | 55.11 | 96.99 | 52.50 | 54.45 | 78.87 | 94.22 | 47.36 | 83.65 | 56.41 |
| DivideMix | 24.69 | 95.75 | 40.94 | 93.32 | 37.02 | 94.36 | 20.81 | 96.10 | 53.10 | 89.35 | 56.39 | 86.71 | 38.82 | 92.60 |
| CM | 17.37 | 96.91 | 21.93 | 95.56 | 17.16 | 96.58 | 39.52 | 92.86 | 30.23 | 93.36 | 61.88 | 84.82 | **31.35** | 93.35 |
| VolMinNet | 13.01 | 97.74 | 15.36 | 97.11 | 14.18 | 97.24 | 60.13 | 80.22 | 45.85 | 87.40 | 55.26 | 86.94 | 33.96 | 91.11 |
| NOODLE | 6.35 | 98.43 | 17.83 | 96.58 | 7.09 | 98.50 | 32.28 | 93.92 | 30.09 | 92.47 | 70.41 | 81.24 | **27.34** | 93.52 |

noise rate at 10%, 30%, and 50% to simulate different levels of noise severity. To test under realistic label noise, we also consider the human-annotated noisy label datasets CIFAR-10N, CIFAR-100N (Wei et al., 2022), and Animal-10N (Song et al., 2019). These are annotated by the crowd workers from the popular crowdsourcing platform Amazon Mechanical Turk (AMT). CIFAR-10N provides five types of noisy label sets: *worst*, *aggregate*, *random1*, *random2*, and *random3*, having 40.21%, 9.03%, 17.23%, 18.12%, and 17.64% overall noise rates, respectively. Animal-10N contains human-annotated labels with noise rate 8%, whereas CIFAR-100N has noise rate of 40.20%. As OOD datasets during test time, we consider several benchmark datasets, including SVHN (Netzer et al., 2011), FashionMNIST (Xiao et al., 2017), LSUN (Yu et al., 2016), iSUN (Pan & Giró-i-Nieto, 2015), Texture (Cimpoi et al., 2013), and Places365 (Zhou et al., 2016).

**Baselines.** We compare our proposed method with several OOD detection baselines as well as different label noise-robust techniques. Regarding the OOD detection baselines, we consider MSP

Table 2: Average OOD detection performance on noisy real datasets; The top two performing algorithms (in terms of average FPR95) are highlighted in bold.

| Method | CIFAR-10N | | | | | | | | | | | | Animal-10N | | CIFAR-100N | |
|---|---|---|---|---|---|---|---|---|---|---|---|---|---|---|---|---|
| | Clean | | Worst | | Agg | | Rand1 | | Rand2 | | Rand3 | | | | | |
| | FPR95 | AUROC | FPR95 | AUROC | FPR95 | AUROC | FPR95 | AUROC | FPR95 | AUROC | FPR95 | AUROC | FPR95 | AUROC | FPR95 | AUROC |
| *OOD baselines* | | | | | | | | | | | | | | | | |
| KNN | 21.06 | 95.80 | 32.48 | 92.89 | 23.95 | 94.84 | 35.48 | 92.65 | 31.99 | 92.70 | 27.27 | 94.09 | 70.44 | 77.04 | 43.20 | 86.54 |
| MSP | 56.43 | 90.07 | 60.15 | 85.49 | 55.04 | 88.21 | 60.75 | 86.62 | 56.44 | 86.50 | 53.90 | 86.70 | 90.64 | 59.90 | 81.08 | 72.66 |
| ODIN | 33.10 | 92.47 | 45.31 | 86.69 | 43.83 | 89.03 | 49.12 | 86.56 | 46.71 | 84.24 | 41.91 | 87.37 | 76.97 | 62.55 | 71.72 | 76.64 |
| Energy | 39.15 | 92.03 | 47.30 | 87.75 | 56.03 | 87.90 | 54.89 | 87.24 | 50.31 | 86.11 | 42.74 | 88.90 | 75.60 | 74.52 | 78.93 | 51.74 |
| ReAct | 60.31 | 83.12 | 65.89 | 78.86 | 47.17 | 91.01 | 68.77 | 76.92 | 65.46 | 79.83 | 57.81 | 81.24 | 79.00 | 71.15 | 76.24 | 67.63 |
| Mahalanobis | 47.22 | 82.91 | 53.57 | 80.61 | 51.14 | 81.76 | 55.26 | 80.05 | 44.25 | 84.60 | 48.37 | 83.37 | 54.54 | 73.00 | 75.15 | 65.47 |
| CIDER | 39.33 | 92.49 | 98.35 | 58.92 | 83.54 | 62.71 | 98.32 | 57.88 | 97.59 | 57.59 | 96.48 | 50.41 | 82.64 | 75.09 | 54.01 | 82.50 |
| SSD+ | 33.66 | 92.42 | 39.69 | 89.82 | 45.72 | 86.07 | 41.90 | 88.77 | 39.37 | 90.52 | 32.40 | 92.23 | 66.11 | 75.28 | 50.29 | 83.36 |
| SNN | 22.60 | 95.53 | **30.87** | 92.78 | 25.14 | 94.18 | 29.87 | 93.76 | 30.74 | 92.94 | 34.26 | 92.12 | 31.43 | 93.65 | **43.15** | 87.13 |
| *Label noise-robust baselines* | | | | | | | | | | | | | | | | |
| SCE | 19.71 | 95.62 | 34.53 | 92.11 | 22.87 | 94.76 | **22.90** | 94.77 | 24.42 | 94.25 | 24.81 | 94.40 | 31.97 | 93.47 | 46.13 | 83.15 |
| GCE | 18.56 | 96.33 | 35.75 | 91.50 | 19.44 | 96.03 | 23.47 | 95.11 | 18.89 | 95.86 | 19.78 | 95.61 | 36.62 | 91.65 | 68.54 | 77.54 |
| DivideMix | 40.81 | 89.16 | 39.32 | 91.53 | 65.83 | 81.83 | 66.64 | 84.52 | 59.27 | 84.22 | 24.81 | 94.40 | 34.27 | 91.77 | 56.28 | 82.92 |
| Co-teaching | 81.94 | 58.15 | 82.14 | 60.68 | 77.98 | 63.71 | 53.42 | 74.23 | 77.59 | 62.76 | 81.94 | 58.15 | 68.47 | 61.72 | 81.68 | 59.61 |
| CM | 18.32 | 96.33 | 36.28 | 89.66 | **21.22** | 95.17 | 24.72 | 94.61 | 23.62 | 95.04 | **20.70** | 95.51 | 33.50 | 92.75 | 49.52 | 85.41 |
| VolMinNet | **15.00** | 96.89 | 37.52 | 91.60 | 23.37 | 94.96 | **22.90** | 95.15 | **18.80** | 96.15 | 22.19 | 95.02 | **29.26** | 94.09 | 56.65 | 81.39 |
| NOODLE | **14.60** | 96.81 | **25.45** | 94.71 | **16.52** | 96.46 | **16.66** | 96.42 | **15.46** | 96.70 | **15.49** | 96.50 | **25.25** | 95.13 | **35.19** | 89.52 |

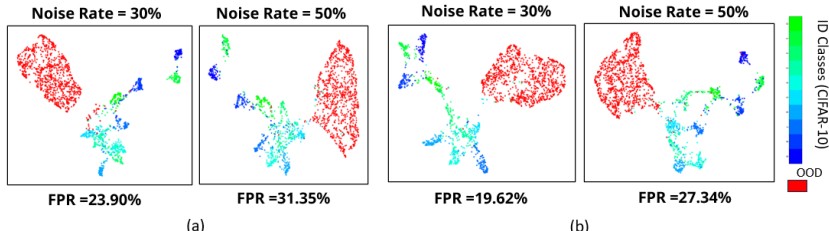

Figure 2: The UMAP representations of the latent feature vectors $h(x)$ learned using (a) CM approach (transition matrix-based label noise correction) with cross-entropy loss, and (b) NOODLE, our proposed approach, on the CIFAR-10 dataset with synthetic label noise.

(Hendrycks & Gimpel, 2017), ODIN (Liang et al., 2018), Energy (Liu et al., 2020), ReAct (Sun et al., 2021), Mahalanobis (Lee et al., 2018), KNN (Sun et al., 2022), CIDER (Ming et al., 2023), SSD+ (Sehwag et al., 2021), and SNN (Ghosal et al., 2024). MSP, ODIN and Energy are softmax-based approaches. MSP relies only on softmax output of the model, while ODIN uses an additional temperature scaling hyperparameter. The Energy method computes an energy-based metric from the model outputs, identifying test samples with higher energy as OOD. ReAct is a logit-based approach. Mahalanobis, KNN, CIDER, SSD+, and SNN are distance-based approaches. The distance-based methods—such as KNN, CIDER, and SNN all advocate the use of non-parametric $k$NN-based score (Sun et al., 2022) for OOD detection.

Regarding label-noise-robust learning methods, we consider different lines of approach for our evaluation. Specifically, we consider CM (Li et al., 2021a), VolMinNet (Li et al., 2021a), SCE (Wang et al., 2019b), GCE (Zhang & Sabuncu, 2018b), DivideMix (Li et al., 2020a), and Co-Teaching (Han et al., 2018b). Here, CM and VolMinNet are probabilistic noise-modeling approaches that rely on transition matrices to correct label noise. GCE and SCE are robust loss function–based approaches that are variants of the cross-entropy loss. DivideMix and Co-Teaching are sample-selection-based approaches that focus on reweighting samples based on the presence of label noise. For OOD performance evaluation, we use the $k$NN-based metric for all these methods, unless specified otherwise.

**Implementation Settings.** We use a CNN-based architecture, DenseNet-101(Huang et al., 2016), as the backbone model for all datasets. We train the model from scratch using the ID datasets. During training for CIFAR-10N and Animal-10N, we set the number of epochs to 100 and use a batch size of 64. First, we extract penultimate layer's features and then apply global average pooling following by $\ell_2$-normalization before performing the PI-based low-rank decomposition module of our NOODLE approach. We initialize the transition matrices as identity matrices of appropriate size in the case of CM-based approaches. We set the number of iterations as $n_{\text{iter}} = 3$ for the PI-based low-rank decomposition in our approach. For all datasets, we choose stochastic gradient

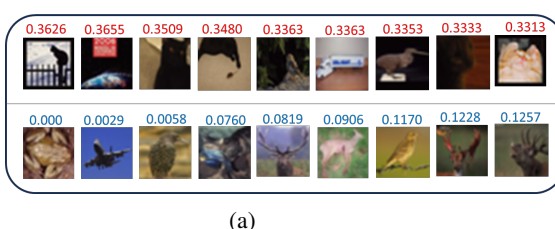
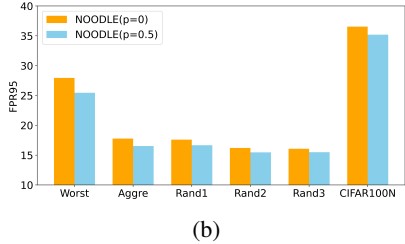

(a)  (b)

Figure 3: (a) Learned "OOD-ness" score $o(\boldsymbol{x}_n)$ for CIFAR-10N training images. Top: top-10 samples with the highest $o(\boldsymbol{x}_n)$ scores. Bottom: bottom-10 samples with the lowest $o(\boldsymbol{x}_n)$ scores. (b) OOD performance of NOODLE with ($p = 0.5$) and without ($p = 0$) robust selection strategy.

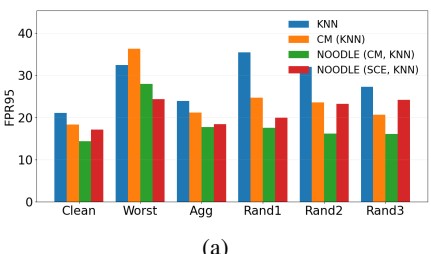
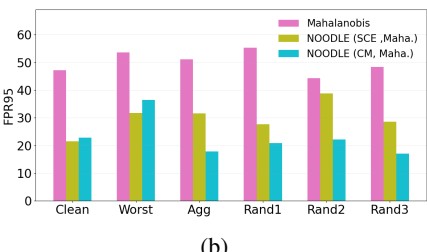

(a)  (b)

Figure 4: Comparison of OOD detection performance (FPR95↓) on CIFAR-10N. (a) Different KNN variants of NOODLE (b) Different Mahalanobis variants of NOODLE. Here "NOODLE (X, Y)" refers to NOODLE with X as loss correction strategy and Y as OOD distance metric.

descent (SGD) as the optimizer with a momentum of 0.9 and a weight decay of $1 \times 10^{-4}$. We tune the hyperparameters $\lambda$ from the set of values $\{0.0001, 0.0005, 0.001, 0.005, 0.1\}$ and $p$ from $\{0, 0.3, 0.5\}$. For the NOODLE approach, we consider different options for loss correction strategies such as CM and SCE. In terms of distance metrics in NOODLE approach, we consider both $k$NN and Mahalanobis scores as OOD detection metrics. We present the best performing variants of the NOODLE approach in the main result tables, yet present the detailed ablation study across different combinations of loss correction and distance metrics in the later sections. We evaluate the OOD detection performance using two widely recognized metrics, i.e., the false positive rate at 95% true positive rate (FPR95) and area under the receiver operating characteristic curve (AUROC). We also present the ID accuracy (ID Acc.) under different settings in the supplementary section in Tables 3-6.

**Results.** Table 1 presents the OOD detection performance of the baselines and our method under symmetric label noise across different noise rates for CIFAR-10 dataset. We can observe that OOD detection baselines that lack label noise-robust training strategy are significantly impacted by high levels of label noise. In contrast, the label noise-robust approaches, especially those based on probabilistic modeling such as CM and VolMinNet maintain relatively strong performance under noisy conditions. Notably, our proposed method, NOODLE, consistently outperforms all other approaches under test in terms of both average FPR95 and AUROC. Our approach is particularly effective at higher noise rates. For example, at 50% noise rate, NOODLE achieves the best performance, with an average FPR95 of 27.34% which represents a reduction in FPR95 of up to 12.5% compared to the best baseline method.

Table 2 presents the OOD detection performance on the real noise datasets which are annotated by unreliable crowd workers. For CIFAR-10N "worst" noise level (the noise rate is about 40.21%), NOODLE achieves a 9.5% reduction in FPR95 and a 1.07% increase in AUROC compared to the best performing baseline SNN. Across other noise settings of CIFAR-10N as well, NOODLE consistently outperforms all baselines. A similar trend is observed on the Animal-10N dataset: while label noise-robust methods such as SCE, CM, and VolMinNet improve FPR95 over other non-robust techniques, NOODLE achieves an additional reduction of approximately 14% compared to the closest baseline. For CIFAR-100N which is more challenging under noisy settings, NOODLE substantially outperforming all the baselines with an improvement of about 15% in FPR95 compared to the best performing baseline. More experiment results are presented in the supplementary section.

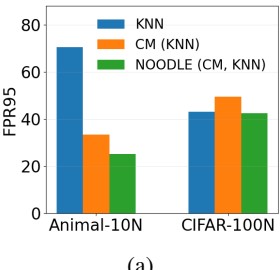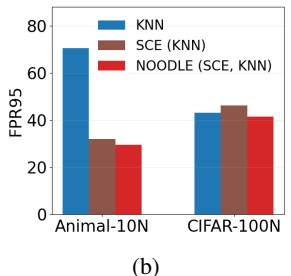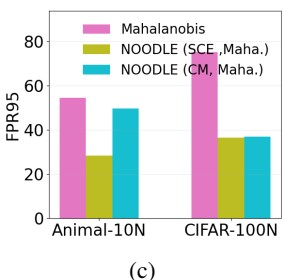

|  (a)  |  (b)  |  (c)  |

Figure 5: Comparison of OOD detection performance (FPR95↓) on Animal-10N and CIFAR-100N datasets. (a) CM vs NOODLE with KNN metric (b) SCE vs NOODLE with KNN metric (c) Different Mahalanobis variants of NOODLE. Here "NOODLE (X, Y)" refers to NOODLE with X as loss correction strategy and Y as OOD distance metric.

**ID vs OOD Features.** From Fig. 1, it is evident that higher noise levels distort the feature space, leading to less compact and more intermixed clusters. This feature distortion degrades the overall OOD detection performance, as we see in our experiments. To illustrate how label noise-robust methods mitigate this issue, we examine the UMAP visualizations in Fig. 2 where the learned features $h(x)$ of the test data for both ID and OOD samples are presented. Here, we compare the feature representations from one of the competing baseline, i.e., CM and our approach NOODLE. For CM, one can observe that cluster distortions are mitigated compared to the scenarios as in Fig. 1, showing that loss correction strategy helps in feature cleaning to some extend. Nonetheless, ID and OOD sample mixups are still present significantly, especially near the boundaries. In contrast, NOODLE produces more compact and well-separated clusters even under high noise rates with much reduced feature mix-up between ID and OOD samples. We also observe the effectiveness of the proposed OOD-ness score in Fig. 3. The "OOD-ness" score $o(x_n)$ clearly differentiates the samples that are more ID-like from those that are not in Fig. 3a. In addition, as shown in Fig. 3b, the average FPR decreases across different real label noise settings when applying the sample selection strategy based on the learned scores $o(x_n)$.

**Ablation Study.** Here, we provide a detailed breakdown of each component of the NOODLE framework by studying the effect of the low-rank and sparse decomposition module in the NOODLE framework under different loss correction strategies and OOD detection metrics. Specifically, we analyze CM and SCE strategies for loss correction, and $k$NN and Mahalanobis scores for the feature distance-based OOD detection. Fig. 4a shows how the NOODLE variant with CM as loss correction and $k$NN as the distance metric substantially advance the OOD detection performance for CIFAR-10N dataset. The SCE variant of the NOODLE version is also reasonably good, yet CM variant (i.e., NOODLE(CM, KNN)) performs much better in all scenarios in CIFAR-10N. For example, in worst case noise version, NOODLE (CM, KNN) reduces FPR95 to 27.94% from 32.48% by KNN, showing its robustness even in challenging settings. We can also observe similar improvement in performance in Fig. 4b, where CM-Mahalanobis variant of the NOODLE also exhibits impressive OOD detection performance across scenarios. These results suggest that under different strategies of loss correction and various OOD detection metrics, the feature cleansing strategy of the NOODLE is effective in improving the ID-OOD separability. We also present similar analysis for other datasets such as Animal-10N and CIFAR-100N in Fig. 5. In CIFAR-100N dataset, the SCE-Mahalanobis variant of NOODLE achieves the best performance, likely because estimating transition matrices for CM-based methods becomes increasingly difficult as the number of classes grows. Nevertheless, our key idea of feature cleaning via low-rank sparse decomposition consistently enhances performance across different settings.

## 5 CONCLUSION

In this work, we introduce a novel framework for OOD detection under noisy labels that addresses the limitations of existing methods by correcting label noise and enhancing OOD performance. By leveraging low-rank ID feature representations, a carefully designed learning criterion, and robust OOD distance metric, our approach provides greater flexibility and effectiveness in improving ID–OOD separability, even in highly noisy settings. Experimental results across multiple benchmarks and challenging OOD scenarios demonstrate the superiority of our method, highlighting its ability to tackle the challenging problem of OOD detection under noisy labels.

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

**Supplementary Material of "Tackling the Noisy Elephant in the Room: Label Noise-robust Out-of-Distribution Detection via Loss Correction and Low-rank Decomposition"**

## A  NOTATION

We use the following notation throughout the paper: $x$, , , and $\mathbf{X}$ represent a scalar, a vector, a matrix, and a tensor, respectively. Both $x_i$ and $[\boldsymbol{x}]_i$ denote the $i$th entry of the vector . $[\boldsymbol{X}]_{i,j}$ denote the $(i,j)$th entry of the matrix $\boldsymbol{X}$. $\boldsymbol{x}_i$ denotes the $i$th row of the matrix $\boldsymbol{X}$; $[I]$ means an integer set $\{1, 2, \ldots, I\}$. denote transpose. $\boldsymbol{X} \geq \mathbf{0}$ implies that all the entries of the matrix $\boldsymbol{X}$ are non-negative. $\mathbb{I}[A]$ denotes an indicator function for the event $A$ such that $\mathbb{I}[A] = 1$ if the event $A$ happens, otherwise $\mathbb{I}[A] = 0$. $\mathsf{CE}(\boldsymbol{x}, y) = -\sum_{k=1}^{K} \mathbb{I}[y = k] \log(\boldsymbol{x}(k))$ denotes the cross entropy function. $\boldsymbol{I}$ denotes an identity matrix of appropriate size. $\mathbf{1}_K$ denotes an all-one vector of size $K$. $|\mathcal{C}|$ denotes the cardinality of the set $\mathcal{C}$. $\Delta^K$ denotes a $(K-1)$-dimensional probability simplex such that $\Delta^K = \{\boldsymbol{u} \in \mathbb{R}^K \,|\, \boldsymbol{u} \geq \mathbf{0}, \mathbf{1}^\top \boldsymbol{u} = 1\}$.

## B  ALGORITHM DESCRIPTION

In this section, we present the NOODLE algorithm. Algorithm 1 provides the complete, step-by-step procedure of our approach using the transition matrix-based loss correction strategy. As discussed earlier, we obtain ID features via a low-rank sparse decomposition. The decomposition routine is detailed in Algorithm 2.

---

**Algorithm 1** Proposed approach NOODLE

**Input:** Noisily labeled data $\{(\boldsymbol{x}_n, \widehat{y}_n)\}_{n=1}^N$, where $\boldsymbol{x}_n \in \mathcal{X}, \widehat{y}_n \in \mathcal{Y}, n_{\text{iter}}$, stopping criterion,$K$ as number of classes
**Output:** Estimated parameters $\boldsymbol{\theta}$ and $\boldsymbol{T}$
1: Initialize Transition Matrix $\boldsymbol{T}$ to identity matrices $\boldsymbol{I}_K$
2: Initialize the parameters $\boldsymbol{\theta}$ of the neural network function class $\mathcal{F}$
3: **while** stopping criterion is not reached **do**
4:    **while** stopping criterion is not reached **do**
5:       Draw a random batch $\mathcal{B}$
6:       $\boldsymbol{H_B} \leftarrow [\,\boldsymbol{h}(\boldsymbol{x}_1), \ldots, \boldsymbol{h}(\boldsymbol{x}_B)\,]$ *// penultimate layer features from batch $\mathcal{B}$ where $B$ denotes the batch size (also see Sec. 3)*
7:       $\boldsymbol{Q} \leftarrow \text{ApproxTopKSingularVectors}(\boldsymbol{H_B}, K, n_{\text{iter}})$
8:       $\boldsymbol{H}_{\text{ID}} \leftarrow (\boldsymbol{Q}\boldsymbol{Q}^\top)\boldsymbol{H_B}$
9:       $\boldsymbol{H}_{\text{OOD}} \leftarrow \boldsymbol{H_B} - \boldsymbol{H}_{\text{ID}}$
10:      Compute $\nabla\mathcal{L}_F(\boldsymbol{T}, \mathcal{B}, \boldsymbol{H}_{OOD}))$
11:      $\boldsymbol{T}, \boldsymbol{\theta} \leftarrow \text{SGDOptimizer}(\boldsymbol{T}, \nabla\mathcal{L}_F(\boldsymbol{T}, \mathcal{B}, \boldsymbol{H}_{\text{OOD}}))$
12:    **end while**
13: **end while**

---

## C  MORE EXPERIMENT RESULTS

In this section, we present more detailed evaluations. While the summary results for CIFAR-10N were reported earlier, we now provide dataset-wise OOD performance along with ID accuracy in Table 3 and Table 4. To ensure fairness, all post-hoc methods are evaluated using the same encoder trained with cross-entropy loss, thereby avoiding any bias in performance analysis. For CIDER and SSD+, we follow prior work but replace their default ResNet-18 encoder with DenseNet-101 for consistency. As a result, these methods may require additional fine-tuning to fully realize their potential. For Animal-10N, the dataset-specific results are reported in Table 5. We find that most baseline methods struggle to achieve a good balance between ID accuracy and OOD detection. In contrast, our proposed method NOODLE delivers consistently strong results across both metrics.

Finally, detailed results on CIFAR-100 are shown in Table 6. As expected, CIFAR-100 is considerably more challenging, leading to significant performance degradation for most baselines. Nevertheless, NOODLE achieves the best OOD detection performance while maintaining a competitive

---

**Algorithm 2** ApproxTopKSingularVectors

---

**Input:** Feature matrix $\boldsymbol{H}_{\mathcal{B}} \in \mathbb{R}^{B \times D}$, target rank $K$, number of iterations $n_{\text{iter}}$
**Output:** Orthonormal matrix $\boldsymbol{Q} \in \mathbb{R}^{D \times K}$ spanning the approximate top-$K$ right singular vectors of $\boldsymbol{H}_{\mathcal{B}}$
1: Randomly initialize $\boldsymbol{Q} \in \mathbb{R}^{D \times K}$    *// D: feature dimension, $K$: target rank*
2: **for** $i = 1$ to $n_{\text{iter}}$ **do**
3:    $\boldsymbol{Z} \leftarrow \boldsymbol{H}_{\mathcal{B}}^{\top}(\boldsymbol{H}_{\mathcal{B}}\boldsymbol{Q})$    *// project $\boldsymbol{Q}$ into column space of $\boldsymbol{H}_{\mathcal{B}}$*
4:    $\boldsymbol{Q} \leftarrow \text{QRDecomposition}(\boldsymbol{Z})$    *// obtain orthonormal basis of $\boldsymbol{Z}$'s column space*
5: **end for**
6: **return** $\boldsymbol{Q}$    *// spans approximate top-$K$ right singular vectors of $\boldsymbol{H}_{\mathcal{B}}$*

---

Table 3: OOD detection performance (FPR95↓ / AUROC↑) on CIFAR-10 under different noise settings using a DenseNet-100 encoder.

| Method | SVHN | | FashionMNIST | | LSUN | | iSUN | | Texture | | Places365 | | Average | | ID Acc. |
|---|---|---|---|---|---|---|---|---|---|---|---|---|---|---|---|
| | FPR95↓ | AUROC↑ | FPR95↓ | AUROC↑ | FPR95↓ | AUROC↑ | FPR95↓ | AUROC↑ | FPR95↓ | AUROC↑ | FPR95↓ | AUROC↑ | FPR95↓ | AUROC↑ | |
| | | | | | | | **Clean** | | | | | | | | |
| KNN | 10.25 | 98.26 | 10.95 | 98.03 | 13.21 | 97.64 | 17.05 | 96.87 | 25.30 | 95.31 | 49.58 | 88.70 | 21.06 | 95.80 | 93.32 |
| MSP | 72.54 | 87.37 | 49.86 | 92.74 | 34.70 | 95.33 | 46.04 | 93.34 | 68.51 | 85.16 | 66.90 | 86.46 | 56.43 | 90.07 | 93.32 |
| ODIN | 55.88 | 89.16 | 16.01 | 97.23 | 3.01 | 99.12 | 8.47 | 98.20 | 60.46 | 82.84 | 54.77 | 88.30 | 33.10 | 92.47 | 93.32 |
| Energy | 73.05 | 87.23 | 15.64 | 97.18 | 4.44 | 98.86 | 23.22 | 96.16 | 67.02 | 83.48 | 51.52 | 89.30 | 39.15 | 92.03 | 93.32 |
| ReAct | 97.03 | 61.00 | 44.59 | 93.52 | 28.01 | 95.84 | 41.26 | 93.42 | 88.14 | 68.86 | 62.85 | 86.07 | 60.31 | 83.12 | 93.32 |
| Mahalanobis | 4.51 | 99.13 | 2.47 | 99.31 | 0.63 | 99.75 | 14.79 | 97.38 | 22.46 | 95.08 | 69.08 | 82.08 | 18.99 | 95.45 | 93.32 |
| CIDER | 18.61 | 96.78 | 27.17 | 95.35 | 31.63 | 94.34 | 42.00 | 93.82 | 56.65 | 89.58 | 59.95 | 85.06 | 39.33 | 92.49 | 90.51 |
| SSD+ | 31.05 | 92.81 | 15.85 | 96.54 | 19.28 | 95.86 | 17.41 | 95.87 | 64.27 | 85.06 | 54.11 | 88.39 | 33.66 | 92.42 | 94.03 |
| SNN | 8.68 | 98.35 | 21.49 | 96.22 | 9.22 | 98.42 | 19.46 | 96.72 | 26.99 | 94.97 | 49.74 | 88.52 | 22.60 | 95.53 | 94.15 |
| SCE | 4.59 | 99.13 | 15.47 | 97.06 | 1.96 | 99.58 | 10.60 | 98.06 | 29.31 | 92.84 | 56.36 | 87.03 | 19.71 | 95.62 | 91.09 |
| GCE | 11.33 | 98.02 | 11.73 | 98.00 | 7.16 | 98.73 | 9.06 | 98.30 | 21.70 | 96.15 | 50.39 | 88.80 | 18.56 | 96.33 | 93.54 |
| DivideMix | 22.35 | 94.72 | 36.90 | 91.74 | 33.41 | 90.87 | 46.05 | 87.62 | 47.11 | 86.55 | 58.04 | 84.45 | 40.81 | 89.16 | 81.22 |
| Co-teaching | 56.55 | 74.88 | 99.90 | 36.86 | 95.18 | 58.31 | 96.28 | 61.89 | 55.83 | 78.99 | 94.03 | 48.08 | 81.94 | 58.15 | 86.22 |
| CM | 8.35 | 98.64 | 8.39 | 98.49 | 7.16 | 98.67 | 15.19 | 97.31 | 19.72 | 96.36 | 51.14 | 88.54 | 18.32 | 96.33 | 94.39 |
| VolMinNet | 2.45 | 99.48 | 13.06 | 97.57 | 5.96 | 98.93 | 6.25 | 98.85 | 15.96 | 97.07 | 46.30 | 89.42 | 15.00 | 96.89 | 94.56 |
| NOODLE | 3.06 | 99.35 | 5.44 | 98.86 | 4.92 | 99.06 | 9.73 | 98.03 | 15.48 | 96.80 | 48.96 | 88.72 | 14.60 | 96.81 | 98.37 |
| | | | | | | | **Noise = Worst** | | | | | | | | |
| KNN | 9.17 | 98.28 | 27.89 | 94.95 | 15.76 | 96.98 | 38.41 | 92.03 | 36.21 | 90.94 | 67.46 | 84.15 | 32.48 | 92.89 | 80.79 |
| MSP | 56.74 | 87.97 | 50.54 | 89.33 | 38.83 | 93.72 | 62.58 | 87.03 | 78.16 | 76.44 | 74.04 | 78.44 | 60.15 | 85.49 | 80.79 |
| ODIN | 45.45 | 90.96 | 33.09 | 93.31 | 13.41 | 97.43 | 32.36 | 93.02 | 71.68 | 72.39 | 75.89 | 73.02 | 45.31 | 86.69 | 80.79 |
| Energy | 37.62 | 93.17 | 35.78 | 93.28 | 14.57 | 97.00 | 47.39 | 90.98 | 76.06 | 74.61 | 72.39 | 77.45 | 47.30 | 87.75 | 80.79 |
| ReAct | 70.94 | 82.60 | 52.84 | 90.05 | 36.40 | 93.87 | 68.31 | 81.53 | 88.92 | 54.21 | 77.90 | 70.87 | 65.89 | 78.86 | 80.79 |
| Mahalanobis | 4.41 | 98.97 | 19.95 | 96.33 | 10.05 | 98.07 | 30.50 | 92.65 | 33.69 | 89.11 | 87.96 | 60.38 | 31.09 | 89.25 | 80.79 |
| CIDER | 99.75 | 58.13 | 98.68 | 66.64 | 99.94 | 52.42 | 99.25 | 61.61 | 93.60 | 65.80 | 98.86 | 48.94 | 98.35 | 58.92 | 80.26 |
| SSD+ | 20.08 | 94.86 | 37.05 | 91.58 | 24.16 | 94.97 | 33.27 | 93.97 | 50.12 | 89.22 | 73.44 | 74.29 | 39.69 | 89.82 | 82.87 |
| SNN | 10.98 | 97.95 | 24.49 | 95.67 | 16.61 | 96.76 | 31.84 | 93.96 | 33.92 | 90.69 | 67.36 | 81.67 | 30.87 | 92.78 | 80.57 |
| SCE | 12.39 | 97.67 | 28.98 | 94.52 | 17.71 | 96.62 | 49.37 | 89.63 | 34.61 | 91.13 | 64.11 | 83.10 | 34.53 | 92.11 | 83.48 |
| GCE | 10.53 | 98.18 | 21.72 | 96.09 | 15.71 | 97.05 | 55.81 | 87.25 | 45.69 | 87.86 | 65.02 | 82.54 | 35.75 | 91.50 | 83.49 |
| DivideMix | 44.70 | 93.52 | 48.20 | 89.20 | 38.09 | 90.29 | 15.35 | 96.70 | 32.06 | 94.42 | 57.95 | 85.09 | 39.39 | 91.53 | 80.23 |
| Co-teaching | 50.04 | 85.37 | 99.95 | 29.04 | 99.61 | 58.63 | 96.88 | 58.36 | 52.15 | 82.65 | 94.22 | 50.03 | 82.14 | 60.68 | 86.34 |
| CM | 8.61 | 98.29 | 17.92 | 96.78 | 28.37 | 90.87 | 56.29 | 81.97 | 39.24 | 88.19 | 67.23 | 81.87 | 36.28 | 89.66 | 76.26 |
| VolMinNet | 7.58 | 98.33 | 42.88 | 90.42 | 19.70 | 95.50 | 43.71 | 89.08 | 40.90 | 87.46 | 70.64 | 77.54 | 37.57 | 89.72 | 76.30 |
| NOODLE | 3.17 | 99.47 | 21.57 | 96.21 | 7.57 | 98.03 | 28.09 | 97.28 | 30.52 | 92.53 | 61.75 | 84.71 | 25.45 | 94.71 | 83.72 |
| | | | | | | | **Noise = Aggre** | | | | | | | | |
| KNN | 11.37 | 97.91 | 18.32 | 96.70 | 9.93 | 98.11 | 18.12 | 96.35 | 23.69 | 95.43 | 62.26 | 84.54 | 23.95 | 94.84 | 89.93 |
| MSP | 57.75 | 89.45 | 47.16 | 90.89 | 27.74 | 95.11 | 59.89 | 88.27 | 67.84 | 83.15 | 69.89 | 82.42 | 55.04 | 88.21 | 89.93 |
| ODIN | 49.75 | 91.54 | 30.78 | 94.69 | 9.29 | 98.25 | 33.50 | 92.13 | 65.18 | 82.01 | 74.50 | 75.54 | 43.83 | 89.03 | 89.93 |
| Energy | 66.66 | 88.00 | 40.23 | 93.66 | 18.07 | 97.10 | 64.95 | 88.41 | 71.46 | 81.96 | 74.80 | 78.26 | 56.03 | 87.90 | 89.93 |
| ReAct | 66.66 | 88.00 | 40.23 | 93.66 | 18.07 | 97.10 | 64.95 | 88.41 | 71.46 | 81.96 | 21.61 | 96.92 | 47.17 | 91.01 | 89.93 |
| Mahalanobis | 4.41 | 98.97 | 19.95 | 96.33 | 10.05 | 98.07 | 30.50 | 92.65 | 33.69 | 89.11 | 87.96 | 60.38 | 31.09 | 89.25 | 89.93 |
| CIDER | 82.88 | 71.63 | 99.61 | 32.85 | 48.08 | 84.89 | 94.50 | 65.86 | 79.66 | 70.30 | 96.50 | 50.70 | 83.54 | 62.71 | 89.06 |
| SSD+ | 63.39 | 76.39 | 34.70 | 90.34 | 23.16 | 94.58 | 43.26 | 90.42 | 36.14 | 90.22 | 73.66 | 74.43 | 45.72 | 86.07 | 90.66 |
| SNN | 6.15 | 98.90 | 14.36 | 97.33 | 8.61 | 98.32 | 35.76 | 92.37 | 26.12 | 94.45 | 59.86 | 83.74 | 25.14 | 94.18 | 90.00 |
| SCE | 4.32 | 99.14 | 19.12 | 96.41 | 4.92 | 98.98 | 19.90 | 96.39 | 24.98 | 94.82 | 63.95 | 82.82 | 22.87 | 94.76 | 90.45 |
| GCE | 1.94 | 99.64 | 11.17 | 98.11 | 7.57 | 98.58 | 14.44 | 97.36 | 28.63 | 94.62 | 52.86 | 87.84 | 19.43 | 96.03 | 91.55 |
| DivideMix | 77.88 | 85.56 | 84.07 | 74.54 | 48.41 | 87.26 | 59.81 | 78.07 | 49.49 | 86.58 | 74.63 | 78.99 | 65.72 | 81.83 | 73.44 |
| Co-teaching | 48.52 | 80.16 | 99.49 | 39.54 | 94.18 | 58.15 | 86.59 | 72.57 | 45.46 | 84.21 | 93.64 | 47.65 | 77.98 | 63.71 | 86.15 |
| CM | 7.82 | 98.68 | 13.01 | 97.55 | 8.41 | 98.30 | 12.70 | 97.62 | 22.45 | 95.18 | 62.91 | 83.70 | 21.22 | 95.17 | 78.82 |
| VolMinNet | 3.88 | 99.17 | 10.28 | 98.08 | 8.46 | 98.32 | 31.84 | 93.84 | 29.41 | 93.06 | 56.36 | 87.27 | 23.37 | 94.96 | 91.86 |
| NOODLE | 1.34 | 99.75 | 17.07 | 97.21 | 2.26 | 98.95 | 8.42 | 98.21 | 17.15 | 96.21 | 52.88 | 88.41 | 16.52 | 96.46 | 91.57 |

and balanced ID accuracy, highlighting its robustness under difficult conditions. We also provide ablation study of hyperparameter $\lambda$ in Table 7. We see the more lower value of $\lambda$ FPR performance is good.Therefore, we choose low value of $\lambda$ in our experimental settings.We present additional ablation study on $K$ in Table 8 and $p$ in Table 10.We also provide additional baselines for CIFAR-10N(worst noise case) in Table 9

Table 4: OOD detection performance (FPR95↓ / AUROC↑) on CIFAR-10 using a DenseNet-100 encoder under **random1**, **random2**, and **random3** noise.

| Method | SVHN | | FashionMNIST | | LSUN | | iSUN | | Texture | | Places365 | | Average | | ID Acc. |
|---|---|---|---|---|---|---|---|---|---|---|---|---|---|---|---|
| | FPR95↓ | AUROC↑ | FPR95↓ | AUROC↑ | FPR95↓ | AUROC↑ | FPR95↓ | AUROC↑ | FPR95↓ | AUROC↑ | FPR95↓ | AUROC↑ | FPR95↓ | AUROC↑ | |
| **Noise = random1** | | | | | | | | | | | | | | | |
| KNN | 26.55 | 95.11 | 32.22 | 94.66 | 19.98 | 96.43 | 36.27 | 92.58 | 30.12 | 93.01 | 67.75 | 84.12 | 35.48 | 92.65 | 88.03 |
| MSP | 63.91 | 89.44 | 56.80 | 88.35 | 33.35 | 94.64 | 70.25 | 83.87 | 70.71 | 81.70 | 69.47 | 81.73 | 60.75 | 86.62 | 88.03 |
| ODIN | 55.83 | 87.93 | 43.60 | 91.41 | 11.17 | 98.01 | 46.12 | 89.32 | 65.07 | 77.53 | 72.94 | 75.16 | 49.12 | 86.56 | 88.03 |
| Energy | 81.76 | 85.45 | 39.75 | 92.56 | 11.32 | 97.78 | 63.37 | 86.34 | 68.55 | 80.04 | 64.57 | 81.25 | 54.89 | 87.24 | 88.03 |
| ReAct | 97.67 | 60.00 | 57.46 | 89.81 | 28.41 | 95.32 | 72.90 | 78.42 | 86.65 | 62.40 | 69.53 | 75.58 | 68.77 | 76.92 | 88.03 |
| Mahalanobis | 16.86 | 96.22 | 48.89 | 88.49 | 13.62 | 97.36 | 50.63 | 82.90 | 36.24 | 87.86 | 92.57 | 50.57 | 43.14 | 83.90 | 88.03 |
| CIDER | 98.03 | 48.59 | 99.91 | 59.70 | 97.94 | 63.79 | 99.45 | 58.63 | 97.02 | 58.86 | 97.60 | 57.74 | 98.32 | 57.88 | 87.74 |
| SSD+ | 18.54 | 93.66 | 51.21 | 87.56 | 20.28 | 95.63 | 36.77 | 91.92 | 51.22 | 89.03 | 73.38 | 74.79 | 41.90 | 88.77 | 90.33 |
| SNN | 16.25 | 97.07 | 35.63 | 93.93 | 15.74 | 97.05 | 23.81 | 95.32 | 26.77 | 93.92 | 61.03 | 85.28 | 29.87 | 93.76 | 87.13 |
| SCE | 17.45 | 97.24 | 10.51 | 97.91 | 5.81 | 98.81 | 14.95 | 97.11 | 25.83 | 93.70 | 62.82 | 83.86 | 22.90 | 94.77 | 89.81 |
| GCE | 6.38 | 98.87 | 11.85 | 97.92 | 11.48 | 97.93 | 23.18 | 96.11 | 31.91 | 93.27 | 56.04 | 86.58 | 23.47 | 95.11 | 90.46 |
| DivideMix | 89.21 | 84.82 | 62.24 | 86.97 | 56.76 | 88.52 | 68.00 | 80.74 | 50.85 | 86.88 | 72.78 | 79.19 | 66.64 | 84.52 | 77.14 |
| Co-teaching | 51.32 | 81.34 | 49.00 | 83.00 | 52.00 | 87.00 | 23.18 | 96.11 | 74.58 | 69.89 | 94.74 | 62.07 | 53.42 | 74.23 | 86.44 |
| CM | 3.13 | 99.35 | 18.50 | 96.55 | 15.74 | 97.22 | 21.04 | 96.18 | 25.85 | 94.65 | 64.08 | 83.69 | 24.72 | 94.61 | 90.31 |
| VolMinNet | 5.48 | 99.01 | 15.74 | 97.11 | 6.92 | 98.75 | 22.86 | 95.86 | 30.30 | 94.07 | 56.10 | 86.06 | 22.90 | 95.15 | 90.53 |
| NOODLE | 3.21 | 99.47 | 15.65 | 97.21 | 5.23 | 98.93 | 5.03 | 98.85 | 12.79 | 97.69 | 58.03 | 86.35 | 16.66 | 96.42 | 90.36 |
| **Noise = random2** | | | | | | | | | | | | | | | |
| KNN | 6.29 | 98.73 | 41.05 | 91.75 | 19.28 | 96.04 | 24.92 | 95.02 | 30.85 | 92.20 | 69.53 | 82.43 | 31.99 | 92.70 | 87.79 |
| MSP | 53.36 | 89.13 | 58.24 | 84.86 | 30.82 | 94.26 | 48.26 | 91.41 | 73.00 | 80.23 | 74.98 | 79.10 | 56.44 | 86.50 | 87.79 |
| ODIN | 44.99 | 87.97 | 56.14 | 83.96 | 13.74 | 97.24 | 19.78 | 94.78 | 68.03 | 73.23 | 77.58 | 68.27 | 46.71 | 84.24 | 87.79 |
| Energy | 60.00 | 87.44 | 50.75 | 87.08 | 17.62 | 96.63 | 29.00 | 93.97 | 71.31 | 76.10 | 73.20 | 75.42 | 50.31 | 86.11 | 87.79 |
| ReAct | 78.12 | 80.74 | 56.58 | 86.77 | 30.49 | 94.12 | 62.46 | 86.31 | 87.27 | 61.33 | 77.87 | 69.69 | 65.46 | 79.83 | 87.79 |
| Mahalanobis | 11.94 | 96.96 | 69.11 | 81.20 | 23.57 | 95.12 | 36.79 | 87.81 | 38.72 | 86.17 | 94.13 | 48.44 | 45.71 | 82.62 | 87.79 |
| CIDER | 96.75 | 67.67 | 99.94 | 45.22 | 99.59 | 46.57 | 99.68 | 67.00 | 91.21 | 60.02 | 98.38 | 59.07 | 97.59 | 57.59 | 87.53 |
| SSD+ | 16.20 | 96.64 | 34.41 | 91.92 | 18.91 | 95.97 | 36.97 | 93.74 | 54.53 | 86.97 | 75.17 | 77.90 | 39.37 | 90.52 | 90.27 |
| SNN | 4.84 | 99.05 | 42.45 | 92.36 | 19.44 | 96.04 | 17.68 | 96.06 | 33.90 | 91.82 | 66.09 | 82.28 | 30.74 | 92.94 | 88.11 |
| SCE | 6.17 | 98.78 | 15.97 | 97.16 | 13.15 | 97.50 | 24.60 | 95.04 | 25.41 | 94.60 | 61.19 | 82.42 | 24.42 | 94.25 | 89.72 |
| GCE | 2.02 | 99.57 | 18.91 | 96.46 | 5.57 | 98.84 | 9.06 | 98.09 | 19.47 | 96.02 | 55.29 | 86.18 | 18.89 | 95.86 | 90.29 |
| DivideMix | 79.28 | 78.13 | 68.73 | 80.40 | 55.05 | 86.76 | 26.58 | 94.70 | 51.37 | 87.91 | 74.59 | 77.40 | 59.27 | 84.22 | 78.30 |
| Co-teaching | 51.32 | 81.34 | 99.87 | 32.07 | 74.58 | 69.89 | 94.74 | 62.07 | 52.39 | 81.73 | 92.63 | 49.46 | 77.59 | 62.76 | 85.99 |
| CM | 1.83 | 99.62 | 16.84 | 97.00 | 17.64 | 96.72 | 18.08 | 96.81 | 27.96 | 94.45 | 59.38 | 85.66 | 23.62 | 95.04 | 90.29 |
| VolMinNet | 3.95 | 99.25 | 15.59 | 97.29 | 8.74 | 98.36 | 8.31 | 98.42 | 22.82 | 95.84 | 53.37 | 87.74 | 18.80 | 96.15 | 94.35 |
| NOODLE | 2.14 | 99.49 | 12.51 | 97.82 | 3.98 | 99.05 | 5.02 | 98.87 | 17.78 | 95.74 | 52.42 | 87.89 | 15.46 | 96.70 | 90.25 |
| **Noise = random3** | | | | | | | | | | | | | | | |
| KNN | 8.79 | 98.41 | 29.45 | 93.81 | 12.45 | 97.66 | 28.16 | 94.63 | 25.80 | 94.26 | 58.99 | 85.76 | 27.27 | 94.09 | 87.77 |
| MSP | 50.92 | 90.23 | 44.39 | 89.13 | 25.79 | 95.34 | 69.78 | 81.85 | 67.02 | 81.86 | 65.51 | 81.76 | 53.90 | 86.70 | 87.77 |
| ODIN | 41.84 | 90.63 | 30.77 | 92.45 | 8.82 | 98.25 | 42.48 | 89.24 | 60.35 | 77.22 | 67.20 | 76.41 | 41.91 | 87.37 | 87.77 |
| Energy | 40.77 | 92.63 | 26.49 | 93.06 | 8.85 | 98.19 | 57.99 | 85.82 | 64.08 | 80.66 | 58.27 | 83.06 | 42.74 | 88.90 | 87.77 |
| ReAct | 77.62 | 77.76 | 34.33 | 92.52 | 16.16 | 96.96 | 67.01 | 79.29 | 85.32 | 62.89 | 66.42 | 77.99 | 57.81 | 81.24 | 87.77 |
| Mahalanobis | 7.87 | 98.31 | 26.49 | 94.43 | 7.01 | 98.64 | 54.91 | 82.04 | 28.95 | 90.66 | 87.15 | 57.60 | 35.40 | 86.95 | 87.77 |
| CIDER+ | 94.37 | 51.43 | 95.95 | 62.08 | 98.10 | 51.80 | 99.37 | 33.13 | 93.19 | 46.85 | 97.89 | 57.20 | 96.48 | 50.41 | 87.16 |
| SSD+ | 31.92 | 93.43 | 29.40 | 93.66 | 12.15 | 97.74 | 25.35 | 93.66 | 36.34 | 91.72 | 59.22 | 83.18 | 32.40 | 92.23 | 90.23 |
| SNN | 12.97 | 97.67 | 34.27 | 91.84 | 16.04 | 96.81 | 53.01 | 89.46 | 29.40 | 92.96 | 59.90 | 83.95 | 34.26 | 92.12 | 87.94 |
| SCE | 6.08 | 98.81 | 16.65 | 96.94 | 13.47 | 96.64 | 24.29 | 95.55 | 23.32 | 95.23 | 65.06 | 83.22 | 24.81 | 94.40 | 89.39 |
| GCE | 7.30 | 98.63 | 17.23 | 96.88 | 5.28 | 98.96 | 10.88 | 97.86 | 20.04 | 95.84 | 57.88 | 85.49 | 19.78 | 95.61 | 90.71 |
| DivideMix | 26.15 | 95.47 | 57.09 | 89.78 | 38.77 | 92.91 | 46.40 | 89.32 | 49.93 | 87.99 | 74.53 | 80.78 | 48.81 | 89.38 | 69.20 |
| Co-teaching | 50.65 | 80.80 | 99.96 | 23.83 | 99.55 | 56.65 | 95.49 | 58.16 | 52.45 | 80.56 | 93.54 | 48.86 | 81.94 | 58.15 | 86.53 |
| CM | 2.56 | 99.48 | 12.00 | 97.75 | 6.82 | 98.60 | 23.46 | 95.99 | 23.74 | 94.94 | 55.65 | 86.32 | 20.70 | 95.51 | 90.79 |
| VolMinNet | 9.80 | 98.06 | 10.10 | 98.02 | 7.61 | 98.40 | 24.25 | 95.63 | 25.74 | 94.40 | 55.65 | 85.58 | 22.19 | 95.02 | 94.36 |
| NOODLE | 2.31 | 99.41 | 12.31 | 97.82 | 3.34 | 99.25 | 5.26 | 98.79 | 17.56 | 95.86 | 52.15 | 87.86 | 15.49 | 96.50 | 90.37 |

Table 5: OOD detection performance (FPR95↓ / AUROC↑) on Animal-10N with real noisy labels using a DenseNet-100 encoder.

| Method | SVHN | | FashionMNIST | | LSUN | | iSUN | | DTD | | Places365 | | Average | | ID Acc. |
|---|---|---|---|---|---|---|---|---|---|---|---|---|---|---|---|
| | FPR95↓ | AUROC↑ | FPR95↓ | AUROC↑ | FPR95↓ | AUROC↑ | FPR95↓ | AUROC↑ | FPR95↓ | AUROC↑ | FPR95↓ | AUROC↑ | FPR95↓ | AUROC↑ | |
| KNN | 45.29 | 89.55 | 58.05 | 90.30 | 66.19 | 79.93 | 90.66 | 62.18 | 73.17 | 74.54 | 89.26 | 65.71 | 70.44 | 77.04 | 81.52 |
| MSP | 93.12 | 64.84 | 82.74 | 79.61 | 84.00 | 74.53 | 96.60 | 34.67 | 96.97 | 41.18 | 90.40 | 64.56 | 90.64 | 59.90 | 81.52 |
| ODIN | 79.29 | 64.67 | 51.20 | 88.26 | 48.52 | 87.84 | 96.52 | 34.97 | 95.43 | 36.66 | 90.85 | 62.89 | 76.97 | 62.55 | 81.52 |
| Energy | 81.91 | 74.54 | 66.32 | 88.64 | 31.94 | 94.83 | 96.32 | 45.16 | 89.02 | 69.24 | 88.09 | 74.71 | 75.60 | 74.52 | 81.52 |
| ReAct | 79.18 | 77.62 | 72.85 | 85.98 | 42.96 | 92.02 | 95.69 | 58.85 | 91.88 | 52.46 | 91.43 | 59.99 | 79.00 | 71.15 | 81.52 |
| Mahalanobis | 31.73 | 91.49 | 97.47 | 42.44 | 68.73 | 66.87 | 4.87 | 98.87 | 33.39 | 86.64 | 91.04 | 51.67 | 54.54 | 73.00 | 81.52 |
| CIDER | 78.00 | 82.28 | 74.75 | 83.81 | 51.20 | 87.60 | 99.00 | 36.84 | 95.13 | 78.33 | 74.75 | 81.70 | 82.64 | 75.09 | 81.85 |
| SSD+ | 62.40 | 82.49 | 59.80 | 84.02 | 40.96 | 87.82 | 97.62 | 36.93 | 76.10 | 78.53 | 59.80 | 81.90 | 66.11 | 75.28 | 82.06 |
| SNN | 41.40 | 91.28 | 11.58 | 97.79 | 12.04 | 97.63 | 39.68 | 91.61 | 29.22 | 93.29 | 54.67 | 90.27 | 31.43 | 93.65 | 81.52 |
| SCE | 29.06 | 90.96 | 41.71 | 93.05 | 27.00 | 94.63 | 9.45 | 98.39 | 29.59 | 94.10 | 55.03 | 89.71 | 31.97 | 93.47 | 81.22 |
| GCE | 28.98 | 91.90 | 43.05 | 91.28 | 23.75 | 94.14 | 27.02 | 95.91 | 30.18 | 92.52 | 66.74 | 84.15 | 36.62 | 91.65 | 80.86 |
| DivideMix | 16.15 | 96.44 | 19.84 | 96.28 | 18.42 | 95.98 | 60.56 | 86.34 | 40.14 | 85.92 | 50.52 | 89.62 | 34.27 | 91.77 | 79.63 |
| Co-teaching | 68.04 | 83.45 | 99.90 | 8.68 | 84.54 | 55.97 | 11.71 | 97.11 | 49.72 | 82.39 | 96.92 | 42.70 | 68.47 | 61.72 | 74.08 |
| CM | 25.84 | 95.25 | 15.41 | 97.22 | 13.42 | 97.44 | 63.61 | 84.11 | 37.06 | 90.88 | 45.65 | 91.58 | 33.50 | 92.75 | 82.48 |
| VolMinNet | 15.08 | 96.79 | 29.22 | 94.51 | 11.80 | 97.73 | 40.62 | 90.67 | 25.20 | 94.64 | 53.63 | 90.21 | 29.26 | 94.09 | 81.78 |
| NOODLE | 26.49 | 94.77 | 24.41 | 95.75 | 11.36 | 97.86 | 17.29 | 96.70 | 18.21 | 96.33 | 53.76 | 89.37 | 25.25 | 95.13 | 82.98 |

Table 6: OOD detection performance (FPR95↓ / AUROC↑) on CIFAR-100 with real noisy labels using a DenseNet-100 encoder.

| Method | SVHN | | FashionMNIST | | LSUN | | iSUN | | DTD | | Places365 | | Average | | ID Acc. |
|---|---|---|---|---|---|---|---|---|---|---|---|---|---|---|---|
| | FPR95↓ | AUROC↑ | FPR95↓ | AUROC↑ | FPR95↓ | AUROC↑ | FPR95↓ | AUROC↑ | FPR95↓ | AUROC↑ | FPR95↓ | AUROC↑ | FPR95↓ | AUROC↑ | |
| KNN | 11.08 | 97.63 | 42.68 | 92.88 | 28.07 | 93.32 | 53.09 | 81.81 | 32.73 | 91.70 | 91.56 | 61.90 | 43.20 | 86.54 | 52.48 |
| MSP | 86.65 | 75.40 | 76.23 | 81.21 | 52.72 | 88.08 | 88.69 | 67.64 | 92.36 | 60.23 | 89.81 | 63.40 | 81.08 | 72.66 | 52.48 |
| ODIN | 93.41 | 68.56 | 58.06 | 87.70 | 26.85 | 95.10 | 70.69 | 83.42 | 92.78 | 59.70 | 88.51 | 65.36 | 71.72 | 76.64 | 52.48 |
| Energy | 97.07 | 47.12 | 53.27 | 91.52 | 35.35 | 93.49 | 90.27 | 68.41 | 99.29 | 3.46 | 98.31 | 6.41 | 78.93 | 51.74 | 52.48 |
| ReAct | 97.07 | 47.12 | 53.27 | 91.52 | 35.35 | 93.49 | 90.27 | 68.41 | 97.53 | 38.90 | 83.94 | 66.32 | 76.24 | 67.63 | 52.48 |
| Mahalanobis | 64.60 | 82.25 | 99.38 | 44.82 | 95.23 | 49.37 | 53.12 | 83.45 | 42.13 | 84.50 | 96.43 | 48.47 | 75.15 | 65.47 | 52.48 |
| CIDER | 28.62 | 94.78 | 14.64 | 97.02 | 66.68 | 77.38 | 73.73 | 72.75 | 57.96 | 80.94 | 82.41 | 72.13 | 54.01 | 82.50 | 57.70 |
| SSD+ | 35.23 | 88.50 | 34.22 | 89.82 | 26.73 | 92.94 | 48.84 | 85.22 | 76.06 | 77.32 | 80.69 | 69.36 | 50.29 | 83.86 | 60.06 |
| SNN | 16.78 | 96.78 | 35.98 | 93.88 | 91.30 | 61.59 | 56.80 | 84.25 | 30.82 | 92.35 | 27.24 | 93.93 | 43.15 | 87.13 | 58.06 |
| SCE | 16.87 | 96.64 | 14.13 | 97.12 | 30.67 | 91.97 | 75.99 | 62.21 | 50.27 | 86.01 | 88.84 | 64.92 | 46.13 | 83.15 | 60.74 |
| GCE | 63.18 | 82.35 | 58.27 | 88.42 | 63.09 | 77.88 | 80.78 | 66.52 | 58.03 | 82.63 | 87.90 | 67.42 | 68.54 | 77.54 | 58.21 |
| DivideMix | 30.53 | 94.00 | 37.22 | 93.63 | 67.81 | 83.31 | 67.09 | 70.49 | 48.55 | 85.01 | 86.47 | 71.10 | 56.28 | 82.92 | 33.26 |
| Co-teaching | 51.77 | 83.17 | 99.96 | 27.69 | 85.42 | 66.48 | 98.34 | 47.85 | 59.41 | 79.01 | 95.20 | 53.47 | 81.68 | 59.61 | 46.18 |
| CM | 35.24 | 92.09 | 40.79 | 92.55 | 36.79 | 90.95 | 54.59 | 79.49 | 40.46 | 89.36 | 89.26 | 68.02 | 49.52 | 85.41 | 59.06 |
| VolMinNet | 36.03 | 91.87 | 50.23 | 91.22 | 45.13 | 88.00 | 74.76 | 61.76 | 43.26 | 88.84 | 90.49 | 66.66 | 56.65 | 81.39 | 60.10 |
| NOODLE | 21.12 | 95.97 | 20.44 | 96.46 | 5.24 | 98.76 | 39.20 | 91.36 | 37.45 | 89.97 | 87.67 | 64.59 | 35.19 | 89.52 | 60.89 |

Table 7: OOD detection performance for different λ values for the "worst" noise case. ID dataset is CIFAR-10 and the encoder architecture is DenseNet-101.

| λ | SVHN | | FashionMNIST | | LSUN | | iSUN | | Texture | | Places365 | | AVG | |
|---|---|---|---|---|---|---|---|---|---|---|---|---|---|---|
| | FPR↓ | AUROC↑ | FPR | AUROC | FPR | AUROC | FPR | AUROC | FPR | AUROC | FPR | AUROC | FPR | AUROC |
| 0.0001 | 3.23 | 99.31 | 23.67 | 95.81 | 8.08 | 98.40 | 36.55 | 93.57 | 32.98 | 91.67 | 63.14 | 83.91 | 27.94 | 93.78 |
| 0.0005 | 19.66 | 95.99 | 42.48 | 90.61 | 25.72 | 94.65 | 18.28 | 95.86 | 33.81 | 91.92 | 59.99 | 82.62 | 33.32 | 91.94 |
| 0.001 | 29.12 | 94.13 | 47.44 | 90.18 | 36.82 | 92.17 | 31.88 | 92.10 | 56.01 | 83.47 | 57.17 | 84.44 | 43.07 | 89.42 |

Table 8: OOD detection performance (FPR95↓ / AUROC↑) using CIFAR-100 with real noisy labels with a DenseNet-100 encoder (varying $K$).

| K | SVHN | | FashionMNIST | | LSUN | | iSUN | | Texture | | Places365 | | AVG | |
|---|---|---|---|---|---|---|---|---|---|---|---|---|---|---|
| | FPR↓ | AUROC↑ | FPR | AUROC | FPR | AUROC | FPR | AUROC | FPR | AUROC | FPR | AUROC | FPR | AUROC |
| 10 | 21.28 | 95.95 | 31.66 | 94.16 | 6.70 | 98.51 | 35.01 | 91.50 | 31.77 | 91.58 | 90.59 | 63.49 | 36.17 | 89.20 |
| 15 | 17.14 | 96.42 | 38.31 | 92.93 | 6.02 | 98.68 | 47.71 | 90.76 | 28.14 | 92.57 | 89.62 | 64.97 | 37.82 | 89.39 |
| 20 | 21.12 | 95.97 | 20.44 | 96.46 | 5.24 | 98.76 | 39.20 | 91.36 | 37.45 | 89.97 | 87.67 | 64.59 | 35.19 | 89.52 |
| 30 | 26.45 | 94.70 | 29.85 | 95.10 | 6.50 | 98.50 | 35.50 | 93.20 | 42.20 | 90.00 | 86.30 | 65.00 | 37.80 | 89.40 |
| 40 | 25.10 | 94.81 | 20.04 | 96.67 | 6.50 | 98.73 | 39.08 | 93.04 | 33.51 | 90.82 | 90.72 | 65.42 | 35.83 | 89.92 |

Table 9: OOD detection performance (FPR95↓ / AUROC↑) on CIFAR-10 with real noisy labels using a DenseNet-100 encoder.

| Method | SVHN | | FashionMNIST | | LSUN | | iSUN | | DTD | | Places365 | | Average | |
|---|---|---|---|---|---|---|---|---|---|---|---|---|---|---|
| | FPR95↓ | AUROC↑ | FPR95↓ | AUROC↑ | FPR95↓ | AUROC↑ | FPR95↓ | AUROC↑ | FPR95↓ | AUROC↑ | FPR95↓ | AUROC↑ | FPR95↓ | AUROC↑ |
| | | | | | | Worst | | | | | | | | |
| FDBD | 48.40 | 91.34 | 70.39 | 75.34 | 60.75 | 81.90 | 69.90 | 73.77 | 74.72 | 72.60 | 95.00 | 38.63 | 69.86 | 72.26 |
| RankFeat | 60.41 | 85.53 | 61.81 | 84.44 | 25.66 | 93.74 | 65.47 | 76.22 | 68.28 | 65.89 | 96.40 | 36.73 | 63.00 | 73.76 |
| NECO | 78.06 | 60.65 | 70.85 | 70.22 | 58.76 | 75.76 | 83.83 | 62.01 | 74.05 | 71.28 | 74.15 | 73.12 | 73.28 | 68.84 |
| NOODLE | 3.17 | 99.47 | 21.57 | 96.21 | 7.57 | 98.03 | 28.09 | 97.28 | 30.52 | 92.53 | 61.75 | 84.71 | 25.45 | 94.71 |
| | | | | | | Agree | | | | | | | | |
| FDBD | 71.83 | 83.66 | 46.31 | 87.52 | 47.19 | 89.26 | 79.11 | 74.58 | 78.30 | 69.04 | 88.81 | 61.12 | 68.59 | 77.53 |
| RankFeat | 24.32 | 94.38 | 48.27 | 91.09 | 14.21 | 97.55 | 70.29 | 77.94 | 61.72 | 74.77 | 92.18 | 57.91 | 51.83 | 82.27 |
| NECO | 81.73 | 66.91 | 66.42 | 72.13 | 65.12 | 67.84 | 46.48 | 82.13 | 76.79 | 67.93 | 61.95 | 79.76 | 66.42 | 72.79 |
| NOODLE | 1.34 | 99.75 | 17.07 | 97.21 | 2.26 | 98.95 | 8.42 | 98.21 | 17.15 | 96.21 | 52.88 | 88.41 | 16.52 | 96.46 |
| | | | | | | Rand1 | | | | | | | | |
| FDBD | 48.40 | 91.34 | 70.39 | 75.34 | 60.75 | 81.90 | 69.90 | 73.77 | 74.72 | 72.60 | 95.00 | 38.63 | 69.86 | 72.26 |
| RankFeat | 40.54 | 89.96 | 78.08 | 74.02 | 30.94 | 91.98 | 93.35 | 48.12 | 69.47 | 65.98 | 97.83 | 32.54 | 68.37 | 67.10 |
| NECO | 87.57 | 55.73 | 50.18 | 82.85 | 53.82 | 80.93 | 68.28 | 78.11 | 70.27 | 77.44 | 64.11 | 81.02 | 65.70 | 76.01 |
| NOODLE | 3.21 | 99.47 | 15.65 | 97.21 | 5.23 | 98.93 | 5.03 | 98.85 | 12.79 | 97.69 | 58.03 | 86.35 | 16.66 | 96.42 |

Table 10: OOD detection under real noisy labels (worst case) on CIFAR-10N for various $p$ values.

| $p$ (%) | SVHN | | FashionMNIST | | LSUN | | iSUN | | Texture | | Places365 | | AVG | |
|---|---|---|---|---|---|---|---|---|---|---|---|---|---|---|
| | FPR ↓ | AUROC ↑ | FPR | AUROC | FPR | AUROC | FPR | AUROC | FPR | AUROC | FPR | AUROC | FPR | AUROC |
| 0 | 13.52 | 97.42 | 29.92 | 94.26 | 12.52 | 97.22 | 28.44 | 94.74 | 29.65 | 92.99 | 63.60 | 83.41 | 29.61 | 93.34 |
| 20 | 8.63 | 98.36 | 30.26 | 94.28 | 12.05 | 97.31 | 17.64 | 96.35 | 27.75 | 93.47 | 63.89 | 83.21 | 26.70 | 93.83 |
| 30 | 8.24 | 98.41 | 30.14 | 94.31 | 11.98 | 97.33 | 17.21 | 96.48 | 27.22 | 93.50 | 64.02 | 83.14 | 26.47 | 93.86 |
| 40 | 7.86 | 98.52 | 30.69 | 94.37 | 12.13 | 97.34 | 15.59 | 96.85 | 26.91 | 93.67 | 65.25 | 83.02 | 26.40 | 93.96 |
| 50 | 3.17 | 99.47 | 21.57 | 96.21 | 7.57 | 98.03 | 28.09 | 97.28 | 30.52 | 92.53 | 61.75 | 84.71 | **25.45** | **94.71** |
| 70 | 5.68 | 98.90 | 30.83 | 94.31 | 12.05 | 97.34 | 13.96 | 97.11 | 26.10 | 93.82 | 67.26 | 82.40 | 25.98 | 93.98 |

