# OpenReview forum: "Tackling the Noisy Elephant in the Room: Label Noise-robust Out-of-Distribution Detection via Loss Correction and Low-rank Decomposition"
_ICLR.cc/2026/Conference — Submitted to ICLR 2026_

### Official Review · Reviewer_kwRU · 2025-10-14

**Soundness:** 3
**Presentation:** 3
**Contribution:** 3
**Rating:** 8
**Confidence:** 4

**Summary:**

This paper studies the problem of OOD detection under noisy labeled datasets. The algorithm utilizes established loss functions that are made for noisy label classification and further regularizes the loss by an intuition on the separability of ID and OOD features. Experiments on training DenseNets from scratch on the synthetic and naturally noisy CIFAR-10 datasets outline successful results.

**Strengths:**

The paper is easy to read and follow. The experimental results of the paper seem strong. The problem is well motivated.

**Weaknesses:**

Although the problem is well-motivated, the solution is not. That is, although the authors reference to Fig 1 to establish their intuition for the effect of noise on OOD performance, I can't follow their argument completely. Mainly that the OOD vs ID classes are generally distinguishable in all the figures.

The authors do not address the added complexity for their regularization and do not mention how much this algorithm increases training time.

The section on 'Low-rank plus Sparse Decomposition' is not well written and cause confusion. The authors need to improve the exposition of this section to clear any confusion.

The tables are hard to understand by themselves and this stems from questionable experimental setups. The tables should have further dividers for methods that are aware of the noisy label setup vs those that are not.

ID accuracies for table 1 are not provided.

The fact that the NOODLE rows are not with a constant loss function or constant score is concerning. That is, the authors seem to experiment with all the noise-aware losses for their NOODLE method and choose the best one to report. This is not a good fair experiment.


The methodology is for training from scratch. Given the prevalence of using pre-trained models, it is perhaps more important these days to look at methods and experiments that work on such pre-trained models.

Moreover, it could be interesting to see results on ViTs as well.

Finally, there are certain baselines that are missing. For example, LINe [1] is a post-hoc method that seems to perform well in many scenarios, at least in the absence of noise. Similarly, there are many works that utilize OOD data during training  like [2] and [3] that could also be interesting to compare with. For a more extensive list please see [https://github.com/Jingkang50/OpenOOD](https://github.com/Jingkang50/OpenOOD).


[1] Ahn YH, Park GM, Kim ST. Line: Out-of-distribution detection by leveraging important neurons. In 2023 IEEE/CVF Conference on Computer Vision and Pattern Recognition (CVPR) 2023 Jun 17 (pp. 19852-19862). IEEE.

[2] Sharifi S, Entesari T, Safaei B, Patel VM, Fazlyab M. Gradient-regularized out-of-distribution detection. InEuropean Conference on Computer Vision 2024 Sep 29 (pp. 459-478). Cham: Springer Nature Switzerland.

[3] Jiang W, Cheng H, Chen M, Wang C, Wei H. Dos: Diverse outlier sampling for out-of-distribution detection. arXiv preprint arXiv:2306.02031. 2023 Jun 3.

**Questions:**

I can not understand some results as they don't seem to make sense. Specifically, Tables 3,4,5, and 6. For example in Table 3, Noise=worst has a noise level of around 40%, yet most algorithms have an accuracy of around 80%. How does this work? Same question for the other tables.

---

> ### Author Response · Authors · 2025-11-26
> **Response to Reviewer kwRU**
>
> We would like to thank the reviewer for your reviewing efforts and insightful comments. Please see our detailed response as follows:
> **[Clarification on how label noise affects ID-OOD separability]** We thank the reviewer for pointing out this. Our goal with Fig. 1 is to illustrate how label noise distorts the ID feature space in a way that harms the kNN-based OOD detection. As the noise rate increases, the ID clusters become less compact and are distorted by mislabeled ID examples that lie in between classes. This causes the learned feature space to lose the ID-ness characteristics, causing many OOD samples to have ID sample neighbours, which directly increases the FPR of kNN-based detectors. This is reflected in Fig. 1 as the noise rate increases, we see more “red” points (OOD)  seeping to the distorted ID feature space, resulting in larger FPR. We will make this clearer in the revised manuscript. To further substantiate this idea, we have also incorporated a histogram of the kNN scores at each noise level and revised the caption of Fig. 1 accordingly.
>
>
>  **[Regarding  “Computational cost”]**
> Thank you for this comment and allowing us to clarify. The added regularization in the NOODLE approach is implemented via a power-iteration (PI)–based low-rank approximation of the batch-wise feature matrix $\boldsymbol{H}$ (see the detailed discussion in Section 3 “Proposed Approach” and Algorithms 1 and 2 in the appendix). Since the PI method uses simple matrix–vector multiplications instead of a full SVD, the additional computational complexity is approximately $\mathcal{O}\big(T(DBK) + DK^2\big),$ where $B$ is the batch size, $K$ is the number of classes (or the chosen rank parameter), $D$ is the feature embedding dimension, and $T$ is the number of PI iterations (set as 3 in our case). This overhead is moderate and substantially cheaper than exact nuclear-norm optimization for low-rank decompositions.
> In experiments, this adds only  about 5–8 ms per batch and the overhead is small compared to the forward/backward passes. We also observe that the full training time of NOODLE is similar to GCE/SCE and 4×–5× faster than DivideMix.
>
> ### Table 1: Training Time Comparison (100 Epochs + Per-Epoch Time)
>
> | Baseline     | Total Time (100 epochs) | Per-Epoch Time |
> |--------------|--------------------------|-----------------|
> | **GCE**       | 1 hr 42 min              | 1.02 min/epoch  |
> | **SCE**       | 55 min                   | 0.55 min/epoch  |
> | **DivideMix** | 5 hr 30 min              | 3.30 min/epoch  |
> | **Co-teaching** | 2 hr 30 min            | 1.50 min/epoch  |
> | **NOODLE**    | 1 hr 30 min              | 0.90 min/epoch  |
>
>
>
> **[Regarding section “Low-rank plus Sparse Decomposition”]** We substantially re-write this section in the latest version of the manuscript. Specifically, we add more intuition, reorganized the presentation of the notation and objective, and explicitly explain the role of each term in the model. We believe this updated version will address the confusion raised by the reviewer.
>
> **[Organization of the result tables]**  Thank you for this helpful suggestion. In the revised manuscript, we will reorganize all main tables to include clear group dividers and headers. We group them as methods that are explicitly designed for (or adapted to) the noisy-label setting, and standard OOD detection baselines that are not noise-aware. We also revised the captions accordingly. In the baseline section of the experiment, we have already detailed each method and its characteristics. Combined with the revised tables, we hope this will address the confusion in the current version.
>
>
>
> **[ID Accuracy Table]** In the original submission, we have already presented the ID accuracies in the appendix section (Table 3,4, and 5). We have already mentioned this in the experiment section as well. Due to space/page limitation, we could not include it in the main tables. We will make this pointer more clearer and explicit in the revised version.

---

> ### Author Response · Authors · 2025-11-26
> **Response to Reviewer kwRU**
>
> **[Evaluating on larger settings]** [Evaluating on larger settings] Thank you for raising this point. As suggested by the reviewer, we extend our evaluation to larger-scale settings using pre-trained models. Specifically, we evaluate the **NOODLE** module on OOD detection tasks under few-shot learning using vision–language models such as **CLIP ViT-B/16** as the visual encoder and **ImageNet-1K** as the in-distribution (ID) dataset. The OOD datasets considered are iNaturalist, SUN, Places, and Texture. As a baseline, we use CoOp [1], which is a cross-entropy–based approach where predicted probabilities are derived from cosine similarities between the image embeddings and text embeddings:
>
>
>
> $\Pr(y = k \mid \mathbf{x}) =$
> $\exp\left( \text{sim}(\mathbf{f}, \mathbf{g}_{k'}) / \tau \right)$
> $/$
>
> $\hspace{3cm} \sum_{k'=1}^{K} \exp\left( \text{sim}(\mathbf{f}, \mathbf{g}_{k'}) / \tau \right)$
>
> Here, $\mathbf{f}$ is the visual embedding of the image $\mathbf{x}$, and $\mathbf{g}_k$ is the textual embedding for class $k$.
>
> We integrate the **NOODLE** module with the same low-rank plus sparse decomposition on the image features as in our other experiments. Both CoOp and NOODLE are fine-tuned under the **16-shot** setting. The results demonstrate that the NOODLE module provides **meaningful improvements** even in these large-scale settings.
>
> ####  Table 2: OOD detection performance on ImageNet-1K under different symmetric noise levels.CLIP is used as the encoder.
>
> | Symmetric Noise | Method | iNaturalist FPR ↓ | iNaturalist AUROC ↑ | SUN FPR ↓ | SUN AUROC ↑ | Places365 FPR ↓ | Places365 AUROC ↑ | Texture FPR ↓ | Texture AUROC ↑ | **Avg FPR ↓** | **Avg AUROC ↑** |
> |:---------------:|:--------|:-----------------:|:-------------------:|:----------:|:-------------:|:----------------:|:------------------:|:-------------:|:----------------:|:--------------:|:----------------:|
> | **30%** | Co-Op  | 20.66 | 95.83 | 27.88 | 92.84 | 36.64 | 89.94 | 52.84 | 84.67 | 34.51 | 90.82 |
> |          | NOODLE | 20.89 | 95.72 | 26.81 | 93.22 | 36.48 | 89.85 | 50.39 | 85.26 | **33.64** | **91.01** |
> | **50%** | Co-Op  | 20.60 | 95.67 | 29.00 | 92.64 | 37.25 | 89.72 | 54.29 | 83.07 | 35.29 | 90.78 |
> |          | NOODLE | 18.30 | 96.08 | 26.60 | 93.61 | 35.82 | 90.48 | 46.52 | 85.71 | **33.81** | **91.47** |
> | **75%** | Co-Op  | 20.43 | 96.19 | 33.89 | 92.17 | 40.16 | 89.77 | 52.32 | 85.27 | 36.70 | 90.85 |
> |          | NOODLE | 20.78 | 95.89 | 28.13 | 92.88 | 38.63 | 90.87 | 52.19 | 85.74 | **34.93** | **91.35** |
>
>
> [1] Zhou, Kaiyang, et al. "Learning to prompt for vision-language models." International Journal of Computer Vision 130.9 (2022): 2337-2348.

---

> ### Author Response · Authors · 2025-11-26
> **Response to Reviewer kwRU**
>
> **[More baselines]** We appreciate the reviewer’s comments and the pointers to additional baselines. As there is a large body of OOD detection work in the literature, it is unfortunately not feasible to include all methods within the scope of our study. Since our focus is specifically on **label-noise–robust OOD detection**, we prioritized the most representative baselines for this setting, i.e., a subset of widely used OOD detectors and noise-robust classification methods that can be naturally adapted for OOD detection.
> For the OOD baselines, we selected state-of-the-art and recent methods that have demonstrated competitive performance and have been peer-reviewed at major AI venues. This choice is also shaped by practical constraints: we evaluate across multiple datasets, several synthetic noise rates, and real noisy-label scenarios, making an exhaustive evaluation across all existing baselines computationally prohibitive.
>
> Nonetheless, in response to the reviewer’s suggestions, we have additionally evaluated three more baselines from the OpenOOD repository, FDBD [2], RankFeat-1 [3], and NECO [4], under various noisy-label scenarios. The corresponding results are reported in the table below.
>
>
> ### Table 3: OOD detection performance (FPR95↓ / AUROC↑) on CIFAR-10 with real noisy labels using a DenseNet-100 encoder.
>
> | Noise Level | Method      | SVHN FPR | SVHN AUROC | FashionMNIST FPR | FashionMNIST AUROC | LSUN FPR | LSUN AUROC | iSUN FPR | iSUN AUROC | Texture FPR | Texture AUROC | Places365 FPR | Places365 AUROC | Avg FPR | Avg AUROC |
> |:-----------:|:------------|:--------:|:-----------:|:-----------------:|:-------------------:|:--------:|:-----------:|:--------:|:------------:|:------------:|:---------------:|:---------------:|:----------------:|:--------:|:-----------:|
> | **Worst**   | FDBD        | 48.40 | 91.34 | 70.39 | 75.34 | 60.75 | 81.90 | 69.90 | 73.77 | 74.72 | 72.60 | 95.00 | 38.63 | 69.86 | 72.26 |
> |             | RankFeat  | 60.41 | 85.53 | 61.81 | 84.44 | 25.66 | 93.74 | 65.47 | 76.22 | 68.28 | 65.89 | 96.40 | 36.73 | 63.00 | 73.76 |
> |             | NECO        | 78.06 | 60.65 | 70.85 | 70.22 | 58.76 | 75.76 | 83.83 | 62.01 | 74.05 | 71.28 | 74.15 | 73.12 | 73.28 | 68.84 |
> |             | **NOODLE**  | **3.17** | **99.47** | **21.57** | **96.21** | **7.57** | **98.03** | **28.09** | **97.28** | **30.52** | **92.53** | **61.75** | **84.71** | **25.45** | **94.71** |
> | **Agree**   | FDBD        | 71.83 | 83.66 | 46.31 | 87.52 | 47.19 | 89.26 | 79.11 | 74.58 | 78.30 | 69.04 | 88.81 | 61.12 | 68.59 | 77.53 |
> |             | RankFeat  | 24.32 | 94.38 | 48.27 | 91.09 | 14.21 | 97.55 | 70.29 | 77.94 | 61.72 | 74.77 | 92.18 | 57.91 | 51.83 | 82.27 |
> |             | NECO        | 81.73 | 66.91 | 66.42 | 72.13 | 65.12 | 67.84 | 46.48 | 82.13 | 76.79 | 67.93 | 61.95 | 79.76 | 66.42 | 72.79 |
> |             | **NOODLE**  | **1.34** | **99.75** | **17.07** | **97.21** | **2.26** | **98.95** | **8.42** | **98.21** | **17.15** | **96.21** | **52.88** | **88.41** | **16.52** | **96.46** |
> | **Rand1**   | FDBD        | 48.40 | 91.34 | 70.39 | 75.34 | 60.75 | 81.90 | 69.90 | 73.77 | 74.72 | 72.60 | 95.00 | 38.63 | 69.86 | 72.26 |
> |             | RankFeat  | 40.54 | 89.96 | 78.08 | 74.02 | 30.94 | 91.98 | 93.35 | 48.12 | 69.47 | 65.98 | 97.83 | 32.54 | 68.37 | 67.10 |
> |             | NECO        | 87.57 | 55.73 | 50.18 | 82.85 | 53.82 | 80.93 | 68.28 | 78.11 | 70.27 | 77.44 | 64.11 | 81.02 | 65.70 | 76.01 |
> |             | **NOODLE**  | **3.21** | **99.47** | **15.65** | **97.21** | **5.23** | **98.93** | **5.03** | **98.85** | **12.79** | **97.69** | **58.03** | **86.35** | **16.66** | **96.42** |
>
>
>
> **[Question about Tables 3,4,5]** Thank you for taking a closer look and for bringing this to attention. The reported results with CIFAR-10 Worst using ResNet-34 is also near 80\%---see the leaderboard on noisy CIFAR-10N http://www.noisylabels.com/. We use the densenet architecture here, which reportedly has higher classification accuracy than ResNet architectures for clean scenarios.
>
> [2] Liu, L. and Qin, Y. Fast decision boundary based out-of-distribution detector. ICML, 2024.
> [3] Yue Song, Nicu Sebe, and Wei Wang. Rankfeat: Rank-1 feature removal for out-of-distribution detection. Advances in Neural Information Processing Systems, 35:17885–17898, 2022.
> [4] Ammar, M. B., Belkhir, N., Popescu, S., Manzanera, A., and Franchi, G. NECO: NEural collapse based out-of-distribution detection. In The Twelfth International Conference on Learning Representations, 2024.

---

### Official Review · Reviewer_v1hQ · 2025-10-23

**Soundness:** 2
**Presentation:** 3
**Contribution:** 2
**Rating:** 2
**Confidence:** 5

**Summary:**

In this work, the authors aim to improve the robustness of OOD detection methods to label noise. To this end, they introduce a novel  NOODLE training framework, consisting of a loss correction module, a decomposition of the pre-logit feature matrix, and a distance-based OOD detector paired with a novel scoring function.

**Strengths:**

The paper is well written and well-structured. The method is clearly motivated. Addressing the robustness of OOD detectors is a potentially important and overlooked task. Observing that robustness to noisy labels might be different for feature-based OOD detectors than for classification tasks is insightful. The paper reports strong gains in the considered setup (more on the setup in weaknesses).

**Weaknesses:**

1. Experimental Setup:
The major weakness of this paper is its experimental setup. Almost all presented experiments are done on (a variant of) Cifar10, with a DenseNet, which is outdated for todays standards. The paper does not consider large-scale setups (ImageNet and beyond), varied pretraining schemes, a range of model architectures (including ViT variants), as is standard in current works (e.g. https://arxiv.org/pdf/2503.08023, https://arxiv.org/abs/2505.18032, …). This is likely connected to the fact that the proposed method is a train-time method that requires training from scratch, and has potentially llimited scalability (more on this below). Further, the selection of OOD detection baselines misses important recent work: (FDBD [1], ViM[2], Maha++ [3], Neco[4], AdaScale[5], RMD[6], NNguide[7], …).

2. Scalability:
I have concerns regarding the scalability of the method. In the text, the authors describe that the proposed method requires decomposing the entire training feature matrix (DxN, with N the total number of training samples). In Algorithm 1, H is also written as the full feature matrix [h(x1), …, h(xN)], but there is also a comment that the features are only from batch B. Therefore it is unclear to me whether the full feature matrix needs to be decomposed for every gradient update or only the feature matrix corresponding to the current batch. In both cases, I suspect significant scalability issues for large datasets or datasets with many classes / large feature dimensions. In the latter case, I also expect dependencies on the batch size.

3. Score Construction:
The authors write that “larger [OOD-ness] scores indicate greater deviation... (e.g., noisy or ambiguous samples), while smaller scores correspond to clean, representative ID examples”. Instead of using the most representative examples, they chose the most challenging ones from the train set (with largest OOD score), and compute the distance to those examples as OOD score. This implies the detector's reference set is built from the noisiest, most ambiguous samples, which is counter-intuitive. This is either an error in the text or a (poorly explained) idea that requires justification - I cannot immediately see why this choice would be “robust”. If the OOD-ness score indeed measures OOD-ness, why not use it directly for OOD detection? It seems to be similar to a residual score, as used in ViM [2] or Neco [4].

4. Hyperparameters:
The method introduces several new hyperparameters (lambda, K, p). Table 7 shows that the method is quite susceptible to changes in lambda (comparing e.g. SVHN and iSUN with lambda = 0.0001 and lambda=0.0005). There is no ablation on K, which likely has a strong impact, especially when transferring to other ID datasets (the mixed results on CIFAR100 indicate this already).


5. More remarks:
- on CIFAR100 NOODLE brings little advantages over just using CM or SCE (Fig 5). Given that CIFAR100 is an extremely similar dataset to CIFAR10, I expect even stronger deviations for other (larger) datasets
- CM and SCE are used but never properly introduced. Overall, the loss correction paragraph is not well explained
- Figure 1: Even though the ID clusters are less separated, there is still a pretty good separation between ID and OOD clusters. It is thus unclear to me why the FPR changes so much.
- The ID accuracy for NOODLE is always higher than for baseline methods (even for clean data there is a difference of 5\%, Table 3), which raises the question if the baseline methods were properly tuned, or NOODLE potentially overfitted by tuning the hyperparameters
- CIDER and SSD+ have suspiciously low ID accuracy (Table 3), and it is surprising that they are so much worse than regular Mahalanobis and KNN


Typos:

359 Energy method (“The” missing)

362 CIDAR instead of CIDER

Algorithm 1: missing ref in line 733, two while statements

225: most columns has (have)

227 do / does

228 solving an optimization problem

[1] Liu, L. and Qin, Y. Fast decision boundary based out-of-distribution detector. ICML, 2024.

[2] Wang, H., Li, Z., Feng, L., and Zhang, W. Vim: Out-of-distribution with virtual-logit matching. In CVPR, 2022.

[3] Müller and Hein, Mahalanobis++: Improving OOD Detection via Feature Normalization, ICML 2025

[4] Ammar, M. B., Belkhir, N., Popescu, S., Manzanera, A., and Franchi, G. NECO: NEural collapse based out-of-distribution detection. In The Twelfth International Conference on Learning Representations, 2024.

[5] Sudarshan Regmi, AdaSCALE: Adaptive Scaling for OOD Detection, arxiv preprint: https://arxiv.org/abs/2503.08023, 2025

[6] Ren, J., Fort, S., Liu, J., Roy, A. G., Padhy, S., and Lakshminarayanan, B. A simple fix to mahalanobis distance for improving near-ood detection, 2021.

[7] Park, J., Jung, Y. G., and Teoh, A. B. J. Nearest neighbor guidance for out-of-distribution detection. In Proceedings of the IEEE/CVF International Conference on Computer Vision

**Questions:**

The core assumption of the method is that the feature matrix H can be decomposed into a low-rank component L and a sparse component S. The sparse component is thought to model “outliers”. However, to me it seems there are two different variants of outliers: ID samples with noisy labels, and OOD samples. Why would S (or H-OOD) capture both noisy labels and OOD samples as a single sparse component in a sensible way?

---

> ### Author Response · Authors · 2025-11-26
> **Response to Reviewer v1hQ**
>
> We thank the reviewer for the careful review of our work and the detailed feedback.
>
>
> **[Large-scale evaluations]** Thank you for the feedback. We chose to closely study the common benchmark datasets such as CIFAR-10 and traditional "training from scratch" approach to carefully analyze the effect of label noise in OOD detection as this topic has rarely been studied in a principled manner in literature. **Even in this classic and widely used setting, our understanding of how label noise affects OOD detection and how to mitigate it remains limited**. For this reason, we adopted this setup as the main experimental framework of our paper. Indeed, this focus led to several valuable findings 1) OOD detection performance is far more sensitive to label noise than ID classification accuracy and **feature space distortions play a significant role in this phenomenon**  2) the state-of-the-art **label-noise–robust methods are often insufficient** to mitigate this degradation. 3) these observations could lead to  more in-depth and careful study, and **our ideas could be effectively extended to larger-scale settings**, including those involving pre-trained large vision–language models and large-scale datasets.
>
> Following the reviewer's suggestion, to provide some preliminary results, we evaluate the effectiveness of NOODLE in large-scale settings. Here, we evaluate the **NOODLE** module on OOD detection tasks using few-shot learning on vision–language models such as **CLIP ViT-B/16** as the visual encoder and **ImageNet-1K** as the ID dataset. The OOD datasets include **iNaturalist**, **SUN**, **Places**, and **Texture**. As a baseline, we use **CoOp** [1], which is a cross-entropy–based loss function where the predicted probabilities are derived from cosine similarities between the image embeddings and text embeddings:
>
>
>
> $\Pr(y = k \mid \mathbf{x}) =$
> $\exp\left( \text{sim}(\mathbf{f}, \mathbf{g}_{k'}) / \tau \right)$
> $/$
>
> $\hspace{3cm} \sum_{k'=1}^{K} \exp\left( \text{sim}(\mathbf{f}, \mathbf{g}_{k'}) / \tau \right)$
>
> Here, $\mathbf{f}$ is the visual embedding of the image $\mathbf{x}$, and $\mathbf{g}_k$ is the textual embedding for class $k$.
>
> We integrate the **NOODLE** module with the same low-rank plus sparse decomposition on the image features as used in our other experiments. Both CoOp and NOODLE are fine-tuned under the **16-shot** setting. The results show that the NOODLE module brings **meaningful improvements** in these large-scale settings as well.
>
>
> ####  Table 1: OOD detection performance on ImageNet-1K under different symmetric noise levels.CLIP is used as the encoder.
>
> | Symmetric Noise | Method | iNaturalist FPR ↓ | iNaturalist AUROC ↑ | SUN FPR ↓ | SUN AUROC ↑ | Places365 FPR ↓ | Places365 AUROC ↑ | Texture FPR ↓ | Texture AUROC ↑ | **Avg FPR ↓** | **Avg AUROC ↑** |
> |:---------------:|:--------|:-----------------:|:-------------------:|:----------:|:-------------:|:----------------:|:------------------:|:-------------:|:----------------:|:--------------:|:----------------:|
> | **30%** | Co-Op  | 20.66 | 95.83 | 27.88 | 92.84 | 36.64 | 89.94 | 52.84 | 84.67 | 34.51 | 90.82 |
> |          | NOODLE | 20.89 | 95.72 | 26.81 | 93.22 | 36.48 | 89.85 | 50.39 | 85.26 | **33.64** | **91.01** |
> | **50%** | Co-Op  | 20.60 | 95.67 | 29.00 | 92.64 | 37.25 | 89.72 | 54.29 | 83.07 | 35.29 | 90.78 |
> |          | NOODLE | 18.30 | 96.08 | 26.60 | 93.61 | 35.82 | 90.48 | 46.52 | 85.71 | **33.81** | **91.47** |
> | **75%** | Co-Op  | 20.43 | 96.19 | 33.89 | 92.17 | 40.16 | 89.77 | 52.32 | 85.27 | 36.70 | 90.85 |
> |          | NOODLE | 20.78 | 95.89 | 28.13 | 92.88 | 38.63 | 90.87 | 52.19 | 85.74 | **34.93** | **91.35** |
>
>
> [1] Zhou, Kaiyang, et al. "Learning to prompt for vision-language models." International Journal of Computer Vision 130.9 (2022): 2337-2348.

---

> ### Author Response · Authors · 2025-11-26
> **Response to Reviewer v1hQ**
>
> **[More baselines]** We appreciate the reviewer’s comments and the pointers to additional baselines. As there are many OOD detection works in the literature, unfortunately we could not include all of them in our study. Nonetheless, in response to the reviewer’s comments, we have evaluated 3 more additional baselines such as FDBD [2], RankFeat-1 [3], and NECO [4] for different noisy scenarios and the results are shown in the below table.
>
> [2] Liu, L. and Qin, Y. Fast decision boundary based out-of-distribution detector. ICML, 2024.
> [3] Yue Song, Nicu Sebe, and Wei Wang. Rankfeat: Rank-1 feature removal for out-of-distribution detection. Advances in Neural Information Processing Systems, 35:17885–17898, 2022.
> [4] Ammar, M. B., Belkhir, N., Popescu, S., Manzanera, A., and Franchi, G. NECO: NEural collapse based out-of-distribution detection. In The Twelfth International Conference on Learning Representations, 2024.
>
>
>
> #### Table 2: OOD detection performance (FPR95↓ / AUROC↑) on CIFAR-10 with real noisy labels using a DenseNet-100 encoder.
>
> | Noise Level | Method      | SVHN FPR | SVHN AUROC | FashionMNIST FPR | FashionMNIST AUROC | LSUN FPR | LSUN AUROC | iSUN FPR | iSUN AUROC | Texture FPR | Texture AUROC | Places365 FPR | Places365 AUROC | Avg FPR | Avg AUROC |
> |:-----------:|:------------|:--------:|:-----------:|:-----------------:|:-------------------:|:--------:|:-----------:|:--------:|:------------:|:------------:|:---------------:|:---------------:|:----------------:|:--------:|:-----------:|
> | **Worst**   | FDBD        | 48.40 | 91.34 | 70.39 | 75.34 | 60.75 | 81.90 | 69.90 | 73.77 | 74.72 | 72.60 | 95.00 | 38.63 | 69.86 | 72.26 |
> |             | RankFeat  | 60.41 | 85.53 | 61.81 | 84.44 | 25.66 | 93.74 | 65.47 | 76.22 | 68.28 | 65.89 | 96.40 | 36.73 | 63.00 | 73.76 |
> |             | NECO        | 78.06 | 60.65 | 70.85 | 70.22 | 58.76 | 75.76 | 83.83 | 62.01 | 74.05 | 71.28 | 74.15 | 73.12 | 73.28 | 68.84 |
> |             | **NOODLE**  | **3.17** | **99.47** | **21.57** | **96.21** | **7.57** | **98.03** | **28.09** | **97.28** | **30.52** | **92.53** | **61.75** | **84.71** | **25.45** | **94.71** |
> | **Agree**   | FDBD        | 71.83 | 83.66 | 46.31 | 87.52 | 47.19 | 89.26 | 79.11 | 74.58 | 78.30 | 69.04 | 88.81 | 61.12 | 68.59 | 77.53 |
> |             | RankFeat  | 24.32 | 94.38 | 48.27 | 91.09 | 14.21 | 97.55 | 70.29 | 77.94 | 61.72 | 74.77 | 92.18 | 57.91 | 51.83 | 82.27 |
> |             | NECO        | 81.73 | 66.91 | 66.42 | 72.13 | 65.12 | 67.84 | 46.48 | 82.13 | 76.79 | 67.93 | 61.95 | 79.76 | 66.42 | 72.79 |
> |             | **NOODLE**  | **1.34** | **99.75** | **17.07** | **97.21** | **2.26** | **98.95** | **8.42** | **98.21** | **17.15** | **96.21** | **52.88** | **88.41** | **16.52** | **96.46** |
> | **Rand1**   | FDBD        | 48.40 | 91.34 | 70.39 | 75.34 | 60.75 | 81.90 | 69.90 | 73.77 | 74.72 | 72.60 | 95.00 | 38.63 | 69.86 | 72.26 |
> |             | RankFeat  | 40.54 | 89.96 | 78.08 | 74.02 | 30.94 | 91.98 | 93.35 | 48.12 | 69.47 | 65.98 | 97.83 | 32.54 | 68.37 | 67.10 |
> |             | NECO        | 87.57 | 55.73 | 50.18 | 82.85 | 53.82 | 80.93 | 68.28 | 78.11 | 70.27 | 77.44 | 64.11 | 81.02 | 65.70 | 76.01 |
> |             | **NOODLE**  | **3.21** | **99.47** | **15.65** | **97.21** | **5.23** | **98.93** | **5.03** | **98.85** | **12.79** | **97.69** | **58.03** | **86.35** | **16.66** | **96.42** |

---

> ### Author Response · Authors · 2025-11-26
> **Response to Reviewer v1hQ**
>
> **[Scalability]** Thank you for this comment and allowing us to clarify. The added regularization in the NOODLE approach is implemented via a power-iteration (PI)–based low-rank approximation of the batch-wise feature matrix $\boldsymbol{H}$ (see the detailed discussion in Section 3 “Proposed Approach” and Algorithms 1 and 2 in the appendix, where we also corrected the typo regarding the feature matrix $\boldsymbol{H}$). Since the PI method uses simple matrix–vector multiplications instead of a full SVD, the additional computational complexity is approximately $\mathcal{O}\big(T(DBK) + DK^2\big),$ where $B$ is the batch size, $K$ is the number of classes (or the chosen rank parameter), $D$ is the feature embedding dimension, and $T$ is the number of PI iterations (set to 3 in our experiments; we will make this explicit in the paper). This overhead is moderate and substantially cheaper than exact nuclear-norm optimization for low-rank decompositions.
>
> In experiments, this adds only  about 5–8 ms per batch and the overhead is small compared to the forward/backward passes. We also observe that the full training time of NOODLE is similar to GCE/SCE and 4×–5× faster than DivideMix.
>
>
> ###  Table 3: Training Time Comparison (100 Epochs + Per-Epoch Time)
>
> | Baseline     | Total Time (100 epochs) | Per-Epoch Time |
> |--------------|--------------------------|-----------------|
> | **GCE**       | 1 hr 42 min              | 1.02 min/epoch  |
> | **SCE**       | 55 min                   | 0.55 min/epoch  |
> | **DivideMix** | 5 hr 30 min              | 3.30 min/epoch  |
> | **Co-teaching** | 2 hr 30 min            | 1.50 min/epoch  |
> | **NOODLE**    | 1 hr 30 min              | 0.90 min/epoch  |
>
>
>
>
> **[Score construction]**  Thank you for raising this point and allowing us to clarify. Indeed, we choose difficult examples from the collection of ID training sets for building a robust distance-based detector. This is an intentional choice in our design. Larger OOD scores are typically attained by hard or boundary ID examples and these are chosen as a critical reference set for distance-based OOD detectors such as kNN. Intuitively, this is similar in spirit to margin-based methods such as SVMs, where the decision boundary is determined by support vectors (the hardest points near the margin). So, if we had chosen the most clean ID representative examples, the OOD detector would be highly conservative, not so robust. This is also reflected in our ablation study presented in Fig. 3 (b). Further expanding our values to more $p$ values, we present the following table:
>
> ###  Table 4: OOD detection performance (FPR95 ↓ / AUROC ↑) **CIFAR-10 with real noisy labels (noise level = ‘worst’)**  using DenseNet-100 encoder  by varying $p$ (%)
>
> | $p$ (%) | SVHN FPR | SVHN AUROC | Fashion FPR | Fashion AUROC | LSUN FPR | LSUN AUROC | iSUN FPR | iSUN AUROC | Texture FPR | Texture AUROC | Places FPR | Places AUROC | **Avg FPR** | **Avg AUROC** |
> |:-----:|:--------:|:-----------:|:------------:|:---------------:|:---------:|:-----------:|:---------:|:------------:|:-------------:|:-------------:|:-------------:|:--------------:|:-----------:|:-------------:|
> | **0**  | 13.52 | 97.42 | 29.92 | 94.26 | 12.52 | 97.22 | 28.44 | 94.74 | 29.65 | 92.99 | 63.60 | 83.41 | **29.61** | **93.34** |
> | **20** | 8.63 | 98.36 | 30.26 | 94.28 | 12.05 | 97.31 | 17.64 | 96.35 | 27.75 | 93.47 | 63.89 | 83.21 | **26.70** | **93.83** |
> | **30** | 8.24 | 98.41 | 30.14 | 94.31 | 11.98 | 97.33 | 17.21 | 96.48 | 27.22 | 93.50 | 64.02 | 83.14 | **26.47** | **93.86** |
> | **40** | 7.86 | 98.52 | 30.69 | 94.37 | 12.13 | 97.34 | 15.59 | 96.85 | 26.91 | 93.67 | 65.25 | 83.02 | **26.40** | **93.96** |
> | **50** | 3.17 | 99.47 | 21.57 | 96.21 | 7.57 | 98.03 | 28.09 | 97.28 | 30.52 | 92.53 | 61.75 | 84.71 | **25.45** | **94.71** |
> | **70** | 5.68 | 98.90 | 30.83 | 94.31 | 12.05 | 97.34 | 13.96 | 97.11 | 26.10 | 93.82 | 67.26 | 82.40 | **25.98** | **93.98** |
>
> **[OOD-ness score for detection]**  In the current approach, OOD-ness score is computed for ID training examples through a joint sparse SVD on on mini-batches of data. As the reviewer pointed out, it is also feasible to directly use the OOD-ness score by first estimating a single global low-dimensional subspace for all ID training data and then computing the residual norm after projecting each test feature to the learned subspace. However, this would require a different (more computationally intensive) algorithmic design and test-time inference than our current approach. Hence, our design choice adopts a hybrid strategy by integrating the OOD-ness score with kNN-like distance metrics, leading to combined benefits from both informative ID sample selection and geometric information by distance scores. In the revised manuscript, we will make this rationale clearer.

---

> ### Author Response · Authors · 2025-11-26
> **Response to Reviewer v1hQ**
>
> **[More ablation studies]** We appreciate the reviewer's feedback. We agree that $\lambda$ has a noticeable influence on performance. This is not entirely surprising as $\lambda$  controls how much suppression is applied to some of the features within the ID dataset itself. Following the reviewer's comments, we have further expanded our ablation studies on $p$ (Table 4) and $K$ (Table 5), showing that NOODLE remains stable across a wide range of values for these hyperparameter configurations.
>
>
> ###  Table 5: OOD detection performance (FPR95 ↓ / AUROC ↑)  using **CIFAR-100 with real noisy labels** with a DenseNet-100 encoder (varying $K$)
>
> | $K$  | SVHN FPR | SVHN AUROC | Fashion FPR | Fashion AUROC | LSUN FPR | LSUN AUROC | iSUN FPR | iSUN AUROC | Texture FPR | Texture AUROC | Places FPR | Places AUROC | **Avg FPR** | **Avg AUROC** |
> |:--:|:--------:|:-----------:|:------------:|:---------------:|:---------:|:-----------:|:---------:|:------------:|:-------------:|:-------------:|:-------------:|:--------------:|:-----------:|:-------------:|
> | **10** | 21.28 | 95.95 | 31.66 | 94.16 | 6.70 | 98.51 | 35.01 | 91.50 | 31.77 | 91.58 | 90.59 | 63.49 | **36.17** | **89.20** |
> | **15** | 17.14 | 96.42 | 38.31 | 92.93 | 6.02 | 98.68 | 47.71 | 90.76 | 28.14 | 92.57 | 89.62 | 64.97 | **37.82** | **89.39** |
> | **20** | 21.12 | 95.97 | 20.44 | 96.46 | 5.24 | 98.76 | 39.20 | 91.36 | 37.45 | 89.97 | 87.67 | 64.59 | **35.19** | **89.52** |
> | **30** | 26.45 | 94.70 | 29.85 | 95.10 | 6.50 | 98.50 | 35.50 | 93.20 | 42.20 | 90.00 | 86.30 | 65.00 | **37.80** | **89.40** |
> | **40** | 25.10 | 94.81 | 20.04 | 96.67 | 6.50 | 98.73 | 39.08 | 93.04 | 33.51 | 90.82 | 90.72 | 65.42 | **35.83** | **89.92** |
>
>
>
> **[Loss Correction section]** We thank the reviewer for his suggestion to improve the readability of the paper. Indeed, we made the discussion brief, giving more weightage to the low-rank sparse decomposition part and robust OOD detector part, since loss correction techniques have been already well-established in the literature. However, we acknowledge that this may have affected the readability of the paper. In the revised version, we update this section to properly introduce CM and SCE. More detailed discussions are relegated to the supplementary section.
>
> **[ID vs OOD separability in Fig 1.]** Thank you for raising this comment and allowing us to clarify. As the noise rate increases, the ID clusters become less compact and are distorted by mislabeled ID examples that lie in between classes. This causes the learned feature space to lose the ID-ness characteristics, causing many OOD samples to have ID sample neighbours, which directly increases the FPR of kNN-based detectors. This is reflected in Fig. 1 as the noise rate increases, we see more “red” points (OOD) seeping to the distorted ID feature space, resulting in larger FPR. We will make this clearer in the revised manuscript. To further substantiate this idea, we have also incorporated a histogram of the kNN scores at each noise level and revised the caption of Fig. 1 accordingly.
>
> **[NOODLE on clean data]** We thank the reviewer for raising this concern. In Table 3, NOODLE and all baseline methods use exactly the same DenseNet backbone and training settings (optimizer, learning rate schedule, number of epochs, etc.). For NOODLE, we keep these settings fixed and only introduce the additional NOODLE-specific hyperparameters (e.g., regularization weights) as described in the implementation details, without extra per-dataset or per-noise-rate tuning.  Note that vanilla cross-entropy training (as used in post-hoc OOD approaches such as KNN, MSP, ODIN etc.) that does not have any additional hyperparameter and achieves around 93.3% accuracy on CIFAR-10 with clean settings. Here, the higher ID accuracy of NOODLE  (around 98%) comes from the additional regularization effect of the proposed low-rank sparse decomposition. To confirm this, we have also included the ID accuracy results on the ResNet-18 model for NOODLE and vanilla CE as below for the clean label setting.
> ###  Table 6: ID accuracy on CIFAR-10 using a ResNet-18 backbone
>
> | Baseline | ID Accuracy (%) |
> |----------|------------------|
> | **CE Loss** | 92.59 |
> | **NOODLE**  | 94.58 |

---

> ### Author Response · Authors · 2025-11-26
> **Response to Reviewer v1hQ**
>
> **[Regarding low id accuracy of CIDER and SSD+]** Thank you for pointing this out. We agree that the ID accuracy of CIDER and SSD+ in Table 3 appears unusually low. In our initial experiments, we followed the public code and README, which recommend running the main scripts directly. However, upon closer inspection now, we realized that these scripts are intended to fine-tune a pretrained classifier, whereas in our setup we were using them to train from scratch. This mismatch which was not clearly mentioned in the README, led to the degraded performance. This was a misconfiguration from our side and not a limitation of these methods.
>
> We have now corrected this by first training the backbone with standard cross-entropy using noisy labels (as for our other baselines), and then applying the official CIDER and SSD+ procedures (using a ResNet-18 backbone with CE loss). Under this properly configured setting, we have the following results compared to NOODLE.  We would like comment that SSD+ approach is particularly effective than CIDER due to its self supervised training strategy using contrastive loss. Yet, NOODLE remains effective in the tested scenarios.
>
>
>
> ### Table 7: OOD detection performance (FPR95↓ / AUROC↑) on CIFAR-10 with symmetric noise using a ResNet-18 encoder.
>
> | Sym Noise | Method | SVHN FPR | SVHN AUROC | Fashion FPR | Fashion AUROC | LSUN FPR | LSUN AUROC | iSUN FPR | iSUN AUROC | Texture FPR | Texture AUROC | Places365 FPR | Places365 AUROC | Avg FPR | Avg AUROC | ID Acc |
> |:---------:|:--------|:---------:|:-----------:|:-------------:|:---------------:|:----------:|:------------:|:----------:|:-------------:|:--------------:|:----------------:|:----------------:|:------------------:|:--------:|:-----------:|:-----:|
> | **30%** | SSD+   | 15.85 | 96.33 | 45.95 | 88.57 | 23.76 | 93.82 | 68.11 | 77.20 | 26.65 | 92.82 | 74.66 | 73.03 | 42.50 | 86.96 | 84.04 |
> |          | CIDER  | 55.41 | 85.28 | 87.20 | 77.56 | 91.62 | 52.81 | 89.04 | 67.42 | 80.76 | 72.57 | 84.72 | 66.85 | 81.46 | 70.41 | 81.58 |
> |          | NOODLE |  8.97 | 98.24 | 23.71 | 95.52 | 15.50 | 96.93 | 60.17 | 85.12 | 48.72 | 88.00 | 55.78 | 86.73 | **35.47** | **91.76** | **89.09** |
> | **50%** | SSD+   | 22.23 | 94.66 | 31.14 | 92.26 | 22.30 | 94.59 | 59.41 | 82.33 | 26.91 | 92.99 | 75.70 | 72.32 | 39.62 | 88.19 | 81.02 |
> |          | CIDER  | 93.23 | 50.19 | 96.39 | 54.93 | 98.25 | 54.32 | 93.67 | 52.30 | 95.12 | 52.09 | 93.35 | 52.90 | 95.00 | 52.79 | 77.23 |
> |          | NOODLE | 36.01 | 93.92 | 33.63 | 93.57 | 15.98 | 96.73 | 32.32 | 93.79 | 42.54 | 90.64 | 60.90 | 85.34 | **36.90** | **92.33** | **85.57** |
>
> In the revised version, we have updated all result tables using the correct procedure for DenseNet backbone and presented these methods in a fair and accurate manner, with a reasonable level of hyperparameter tuning given our time constraints.
>
> **[Typos]** Thank you paying close attention. We have fixed these. We will also do a thorough pass to make sure there are no typos and formatting issues in the revised version.
>
> **[Question regarding the $\boldsymbol{H_{OOD}}$ matrix]** We thank the reviewer for this insightful discussion. Note that during training we do not have access to any OOD sample explicitly. As the reviewer points out, the outlier content captured in **$\boldsymbol{H_{OOD}}$ matrix during training can include both spurious features learned due to the influence of noisily labeled training examples as well as the feature components within the correctly labeled training samples that deviate from ID characteristics**. In our method, we do not attempt to distinguish these components explicitly. Instead, $\boldsymbol{H_{OOD}}$ is designed to capture anything that is non-ID, while the low-rank component focuses on learning the cleanest possible version of the ID feature space. This is intuitive since our primary training goal is not to differentiate different $\boldsymbol{H_{OOD}}$ components, but to learn a robust ID representation.  To add to this discussion, we also notice that all the training images with large “OOD-ness” score  in Fig. 3 of the manuscript are labeled incorrectly in the CIFAR-10N dataset, which makes sense as visually ambiguous images or confusing ID examples are more prone to get annotated with noisy labels by human labelers. We will clarify this interpretation in the revised manuscript.

---

### Official Review · Reviewer_udQU · 2025-10-31

**Soundness:** 2
**Presentation:** 2
**Contribution:** 2
**Rating:** 4
**Confidence:** 4

**Summary:**

This paper proposes the *NOODLE* framework, which purifies feature embeddings by integrating loss correction with low-rank sparse decomposition. And one OOD score is introduced to quantify "OOD-ness" content in training data, thereby computing robust OOD detection scores. Extensive experiments demonstrate that NOODLE achieves strong OOD detection performance when the training data contains noise.

**Strengths:**

1. OOD detection is important for safety-critic applications of machine learning models.
2. The setting of OOD detection under noisy training labels is both practical and interesting.
3. Extensive experiments demonstrate the efficacy of the method.

**Weaknesses:**

1. The effectiveness of NOODLE: Since NOODLE requires SVD to compute singular vectors for the Low-rank plus Sparse Decomposition, its computational efficiency is still a concern. Although the authors claim the acceleration via power iteration, the computational cost of this method is still very high due to the typically high dimensionality of the feature matrix.
2. Compatibility issues with logit-based methods: The paper's experimental setup does not include testing logit-based post-hoc methods (e.g., MSP) on the fine-tuned model, as it only uses KNN as the post-hoc score. Therefore, it remains an open question whether this fine-tuning degrades the performance of logit-based methods.
3. Lack of OOD detection evaluation on the clean-label setting: While the paper's primary focus is on the noisy-label training setting, this setting's practical applicability is limited by the difficulty of assessing noise levels in the real world. (We commend the authors for their setup of different noise levels in the main experiments.) The paper would be more solid if it also included an evaluation on clean training datasets. This would verify the score's baseline effectiveness in a conventional setting and enhance the paper's completeness.
4. Lack of comprehensive experiments: The experiments were performed with only CNN architecture (The inclusion of experiments on ViT would strengthen the paper's claims). And no experiments were carried out on the ImageNet benchmark, which is more challenging for OOD detection.
5. Lack of comprehensive comparison: A number of recent related works have not been compared in the paper, e.g., [1], [2], and [3].
6. Questions Regarding Some Test Results: As shown in Table 3 of the Appendix, the ID accuracy of CIDER and SSD+ is particularly low, which suggests that its hyper-parameters may not have been properly tuned.

[1] Haoqi Wang, Zhizhong Li, Litong Feng, and Wayne Zhang.Vim: Out-of-distribution with virtual-logit matching. In Proceedings of the IEEE/CVF Conference on Computer Vision and Pattern Recognition, 2022.

[2] Yue Song, Nicu Sebe, and Wei Wang. Rankfeat: Rank-1 feature removal for out-of-distribution detection. Advances in Neural Information Processing Systems, 35:17885–17898, 2022.

[3] Yue Song, Wei Wang, and Nicu Sebe. Rankfeat&rankweight: Rank-1 feature/weight removal for out-of-distribution detection. IEEE Transactions on Pattern Analysis and Machine Intelligence, 2024.

**Questions:**

Please see the Weaknesses.

---

> ### Author Response · Authors · 2025-11-26
> **Response to Reviewer udQU**
>
> We thank the reviewer for appreciating our idea and for providing constructive feedback. Please see our detailed response as follows:
>
>
> **[Regarding computational cost]** Thank you for the comment. Although NOODLE requires low-rank decomposition, we avoid full SVD and use a lightweight power-iteration step on mini-batch features (also see the detailed discussion in Section 3 “Proposed Approach” and Algorithm 1 and 2 in the appendix section). That is, the PI method uses simple matrix–vector multiplications instead of direct SVD, the added complexity is about $\mathcal{O}\big(T(DBK) + DK^2\big),$ where $B$ is the batch size, $K$ is the number of classes (or the rank parameter set), $D$ is the feature embedding dimension, and $T$ is the number of PI iterations (set as 3 in our case). This overhead is moderate and much cheaper than exact nuclear-norm optimization for low-rank decompositions.
> In experiments, this adds only about 0.9 min/epoch and the overhead is small compared to the forward/backward passes. We also observe that the full training time of NOODLE is similar to GCE/SCE and 4–5 times faster than DivideMix.
>
>
> ###  Table 1: Training Time Comparison (100 Epochs + Per-Epoch Time)
>
> | Baseline     | Total Time (100 epochs) | Per-Epoch Time |
> |--------------|--------------------------|-----------------|
> | **GCE**       | 1 hr 42 min              | 1.02 min/epoch  |
> | **SCE**       | 55 min                   | 0.55 min/epoch  |
> | **DivideMix** | 5 hr 30 min              | 3.30 min/epoch  |
> | **Co-teaching** | 2 hr 30 min            | 1.50 min/epoch  |
> | **NOODLE**    | 1 hr 30 min              | 0.90 min/epoch  |
>
>
> **[Compatibility issues with logit-based methods]** Thank you for raising this point. To address the reviewer’s concerns, we applied MSP in the inference stage after training under NOODLE. Across the three noise settings, MSP on NOODLE achieves average FPR reductions of 13–20% and AUROC improvements of 4–7 points compared to MSP on the baseline model (trained using CE without NOODLE). This shows that fine-tuning with NOODLE does not harm logit-based scores, but enhances them quite significantly.
>
> ###  Table 2: OOD Detection Results (MSP vs NOODLE with MSP posthoc score under different noise levels on CIFAR-10N)
>
> | Noise Level | Method | SVHN FPR | SVHN AUROC | Fashion FPR | Fashion AUROC | LSUN FPR | LSUN AUROC | iSUN FPR | iSUN AUROC | Texture FPR | Texture AUROC | Places365 FPR | Places365 AUROC | Avg FPR | Avg AUROC |
> |------------|---------|----------|------------|--------------|----------------|-----------|-------------|-----------|-------------|--------------|----------------|----------------|------------------|---------|-----------|
> | **Worst** | MSP | 56.74 | 87.97 | 50.54 | 89.33 | 38.83 | 93.72 | 62.58 | 87.03 | 78.16 | 76.44 | 74.04 | 78.44 | 60.15 | 85.49 |
> | | NOODLE (MSP Score) | 72.07 | 88.35 | 17.56 | 96.99 | 8.80 | 98.27 | 61.83 | 90.56 | 61.76 | 84.44 | 59.77 | 86.03 | **46.96** | **90.77** |
> | **Agree** | MSP | 57.55 | 89.45 | 47.16 | 90.89 | 27.74 | 95.11 | 59.89 | 88.27 | 67.84 | 83.15 | 69.89 | 82.42 | 55.04 | 88.21 |
> | | NOODLE (MSP Score) | 49.09 | 93.03 | 13.88 | 97.52 | 7.84 | 98.37 | 37.46 | 94.26 | 54.52 | 88.58 | 55.68 | 88.12 | **36.41** | **93.31** |
> | **Rand1** | MSP | 63.91 | 89.44 | 56.80 | 88.35 | 33.35 | 94.64 | 70.25 | 83.87 | 70.71 | 81.70 | 69.47 | 81.73 | 60.75 | 86.62 |
> | | NOODLE (MSP Score) | 63.28 | 88.14 | 13.89 | 97.41 | 8.37 | 98.35 | 30.70 | 95.54 | 60.41 | 86.14 | 53.21 | 88.66 | **38.31** | **92.37** |

---

> ### Author Response · Authors · 2025-11-26
> **Response to Reviewer udQU**
>
> **[NOODLE’s performance in clean-label setting]** Thank you for the suggestion. We note that Table 3 in the appendix already reports the clean-label setting on the CIFAR-10 dataset, where models are trained without any label noise. Due to page limitations, we chose this organization in the original version. For your reference, we present a subset of these results here, comparing NOODLE with the top baselines in the clean setting.
>
> ###  Table 3: OOD Detection Under Clean Training
>
> | Method      | SVHN FPR | SVHN AUROC | Fashion FPR | Fashion AUROC | LSUN FPR | LSUN AUROC | iSUN FPR | iSUN AUROC | Texture FPR | Texture AUROC | Places365 FPR | Places365 AUROC | Avg FPR | Avg AUROC |
> |-------------|-----------|-------------|--------------|----------------|-----------|-------------|-----------|-------------|--------------|----------------|----------------|------------------|----------|------------|
> | **KNN**       | 10.25 | 98.26 | 10.95 | 98.03 | 13.21 | 97.64 | 17.05 | 96.87 | 25.30 | 95.31 | 49.58 | 88.70 | 21.06 | 95.80 |
> | **SCE**       | 4.59  | 99.13 | 15.47 | 97.06 | 1.96  | 99.58 | 10.60 | 98.06 | 29.31 | 92.84 | 56.36 | 87.03 | 19.71 | 95.62 |
> | **CM**        | 8.35  | 98.64 | 8.39  | 98.49 | 7.16  | 98.67 | 15.19 | 97.31 | 19.72 | 96.36 | 51.14 | 88.54 | 18.32 | 96.33 |
> | **VolMinNet** | 2.45  | 99.48 | 13.06 | 97.57 | 5.96  | 98.93 | 6.25  | 98.85 | 15.96 | 97.07 | 46.30 | 89.42 | 15.00 | 96.89 |
> | **NOODLE**    | 3.06  | 99.35 | 5.44  | 98.86 | 4.92  | 99.06 | 9.73  | 98.03 | 15.48 | 96.80 | 48.96 | 88.72 | **14.60** | **96.81** |
>
> As shown, NOODLE remains competitive even under clean training, further demonstrating its reliability in practical scenarios where it is unclear whether the dataset is noisy.
>
>
> **[Large-scale evaluations]** Thank you for raising this valid comment. To address this, we additionally include results using a pre-trained vision transformer–based encoder (CLIP ViT-B/16) with ImageNet-1K as the ID dataset under symmetric label noise. The OOD datasets considered are iNaturalist, SUN, Places, and Texture. As a baseline, we use CoOp [2], which is a cross-entropy–based loss function where predicted probabilities are derived from cosine similarities between the image embeddings and text embeddings:
>
> $\Pr(y = k \mid \mathbf{x}) =$
> $\exp\left( \text{sim}(\mathbf{f}, \mathbf{g}_{k'}) / \tau \right)$
> $/$
>
> $\hspace{3cm} \sum_{k'=1}^{K} \exp\left( \text{sim}(\mathbf{f}, \mathbf{g}_{k'}) / \tau \right)$
>
>
>
> Here, $\mathbf{f}$ is the visual embedding of the image $\mathbf{x}$, and $\mathbf{g}_k$ is the textual embedding for class $k$.
>
> We integrate the NOODLE module with the same low-rank plus sparse decomposition on the image features as in our other experiments. Both CoOp and NOODLE are fine-tuned under the 16-shot setting. The results show that the NOODLE module brings meaningful improvements in these large-scale settings as well.
>
>
> ###  Table 4: OOD detection performance on ImageNet-1K under different symmetric noise levels. CLIP is used as the encoder.
>
> | Symmetric Noise | Method | iNaturalist FPR ↓ | iNaturalist AUROC ↑ | SUN FPR ↓ | SUN AUROC ↑ | Places365 FPR ↓ | Places365 AUROC ↑ | Texture FPR ↓ | Texture AUROC ↑ | **Avg FPR ↓** | **Avg AUROC ↑** |
> |:---------------:|:--------|:-----------------:|:-------------------:|:----------:|:-------------:|:----------------:|:------------------:|:-------------:|:----------------:|:--------------:|:----------------:|
> | **30%** | Co-Op  | 20.66 | 95.83 | 27.88 | 92.84 | 36.64 | 89.94 | 52.84 | 84.67 | 34.51 | 90.82 |
> |          | NOODLE | 20.89 | 95.72 | 26.81 | 93.22 | 36.48 | 89.85 | 50.39 | 85.26 | **33.64** | **91.01** |
> | **50%** | Co-Op  | 20.60 | 95.67 | 29.00 | 92.64 | 37.25 | 89.72 | 54.29 | 83.07 | 35.29 | 90.78 |
> |          | NOODLE | 18.30 | 96.08 | 26.60 | 93.61 | 35.82 | 90.48 | 46.52 | 85.71 | **33.81** | **91.47** |
> | **75%** | Co-Op  | 20.43 | 96.19 | 33.89 | 92.17 | 40.16 | 89.77 | 52.32 | 85.27 | 36.70 | 90.85 |
> |          | NOODLE | 20.78 | 95.89 | 28.13 | 92.88 | 38.63 | 90.87 | 52.19 | 85.74 | **34.93** | **91.35** |
>
> [1] Zhou, Kaiyang, et al. "Learning to prompt for vision-language models." International Journal of Computer Vision 130.9 (2022): 2337-2348.

---

> ### Author Response · Authors · 2025-11-26
> **Response to Reviewer udQU**
>
> **[More Baselines]** We appreciate the reviewer’s comments and the pointers to additional baselines. As there are many OOD detection works in the literature, unfortunately we could not include all of them in our study. Since our focus is specifically on label-noise robust OOD detection, we have prioritized the most representative baselines in this setting, i.e., a subset of widely used OOD detectors and noise-robust classification methods that can be naturally adapted to OOD detection. This choice is also due to practical constraints as we evaluate across multiple datasets, several synthetic noise rates, and real noisy-label scenarios, which makes running a very large number of baselines computationally expensive. Nonetheless, in response to the reviewer’s comments, we have evaluated 3 more additional baselines such as FDBD [2], RankFeat-1 [3], and NECO [4] for different noisy scenarios and the results are shown in the below table.
>
>
>  ### Table 5: OOD detection performance (FPR95↓ / AUROC↑) on CIFAR-10 with real noisy labels using a DenseNet-100 encoder.
>
> | Noise Level | Method      | SVHN FPR | SVHN AUROC | FashionMNIST FPR | FashionMNIST AUROC | LSUN FPR | LSUN AUROC | iSUN FPR | iSUN AUROC | Texture FPR | Texture AUROC | Places365 FPR | Places365 AUROC | Avg FPR | Avg AUROC |
> |:-----------:|:------------|:--------:|:-----------:|:-----------------:|:-------------------:|:--------:|:-----------:|:--------:|:------------:|:------------:|:---------------:|:---------------:|:----------------:|:--------:|:-----------:|
> | **Worst**   | FDBD        | 48.40 | 91.34 | 70.39 | 75.34 | 60.75 | 81.90 | 69.90 | 73.77 | 74.72 | 72.60 | 95.00 | 38.63 | 69.86 | 72.26 |
> |             | RankFeat  | 60.41 | 85.53 | 61.81 | 84.44 | 25.66 | 93.74 | 65.47 | 76.22 | 68.28 | 65.89 | 96.40 | 36.73 | 63.00 | 73.76 |
> |             | NECO        | 78.06 | 60.65 | 70.85 | 70.22 | 58.76 | 75.76 | 83.83 | 62.01 | 74.05 | 71.28 | 74.15 | 73.12 | 73.28 | 68.84 |
> |             | **NOODLE**  | **3.17** | **99.47** | **21.57** | **96.21** | **7.57** | **98.03** | **28.09** | **97.28** | **30.52** | **92.53** | **61.75** | **84.71** | **25.45** | **94.71** |
> | **Agree**   | FDBD        | 71.83 | 83.66 | 46.31 | 87.52 | 47.19 | 89.26 | 79.11 | 74.58 | 78.30 | 69.04 | 88.81 | 61.12 | 68.59 | 77.53 |
> |             | RankFeat  | 24.32 | 94.38 | 48.27 | 91.09 | 14.21 | 97.55 | 70.29 | 77.94 | 61.72 | 74.77 | 92.18 | 57.91 | 51.83 | 82.27 |
> |             | NECO        | 81.73 | 66.91 | 66.42 | 72.13 | 65.12 | 67.84 | 46.48 | 82.13 | 76.79 | 67.93 | 61.95 | 79.76 | 66.42 | 72.79 |
> |             | **NOODLE**  | **1.34** | **99.75** | **17.07** | **97.21** | **2.26** | **98.95** | **8.42** | **98.21** | **17.15** | **96.21** | **52.88** | **88.41** | **16.52** | **96.46** |
> | **Rand1**   | FDBD        | 48.40 | 91.34 | 70.39 | 75.34 | 60.75 | 81.90 | 69.90 | 73.77 | 74.72 | 72.60 | 95.00 | 38.63 | 69.86 | 72.26 |
> |             | RankFeat  | 40.54 | 89.96 | 78.08 | 74.02 | 30.94 | 91.98 | 93.35 | 48.12 | 69.47 | 65.98 | 97.83 | 32.54 | 68.37 | 67.10 |
> |             | NECO        | 87.57 | 55.73 | 50.18 | 82.85 | 53.82 | 80.93 | 68.28 | 78.11 | 70.27 | 77.44 | 64.11 | 81.02 | 65.70 | 76.01 |
> |             | **NOODLE**  | **3.21** | **99.47** | **15.65** | **97.21** | **5.23** | **98.93** | **5.03** | **98.85** | **12.79** | **97.69** | **58.03** | **86.35** | **16.66** | **96.42** |
>
> [2] Liu, L. and Qin, Y. Fast decision boundary based out-of-distribution detector. ICML, 2024.
> [3] Yue Song, Nicu Sebe, and Wei Wang. Rankfeat: Rank-1 feature removal for out-of-distribution detection. Advances in Neural Information Processing Systems, 35:17885–17898, 2022.
> [4] Ammar, M. B., Belkhir, N., Popescu, S., Manzanera, A., and Franchi, G. NECO: NEural collapse based out-of-distribution detection. In The Twelfth International Conference on Learning Representations, 2024.

---

> ### Author Response · Authors · 2025-11-26
> **Response to Reviewer udQU**
>
> **[Questions about CIDER and SSD+]** Thank you for pointing this out and bringing to our attention. We agree that the ID accuracy of CIDER and SSD+ in Table 3 appears unusually low. In our initial experiments, we followed the public code and README, which recommend running the main scripts directly. However, upon closer inspection now, we realized that these scripts are intended to fine-tune a pretrained classifier, whereas in our setup we were using them to train from scratch. This mismatch which was not clearly mentioned in the README, led to the degraded performance. This was a misconfiguration from our side and not a limitation of these methods.
>
> We have now corrected this by first training the backbone with standard cross-entropy using noisy labels (as for our other baselines), and then applying the official CIDER and SSD+ procedures (using a ResNet-18 backbone with CE loss). Under this properly configured setting, we have the following results compared to NOODLE. In the revised version, we have updated all result tables using the correct procedure for DenseNet backbone and presented these methods in a fair and accurate manner, with a reasonable level of hyperparameter tuning given our time constraints.
>
>
>
> ### Table: OOD detection performance (FPR95↓ / AUROC↑) on CIFAR-10 with symmetric noise using a ResNet-18 encoder.
>
> | Sym Noise | Method | SVHN FPR | SVHN AUROC | Fashion FPR | Fashion AUROC | LSUN FPR | LSUN AUROC | iSUN FPR | iSUN AUROC | Texture FPR | Texture AUROC | Places365 FPR | Places365 AUROC | Avg FPR | Avg AUROC | Acc |
> |:---------:|:--------|:---------:|:-----------:|:-------------:|:---------------:|:----------:|:------------:|:----------:|:-------------:|:--------------:|:----------------:|:----------------:|:------------------:|:--------:|:-----------:|:-----:|
> | **30%** | SSD+    | 15.85 | 96.33 | 45.95 | 88.57 | 23.76 | 93.82 | 68.11 | 77.20 | 26.65 | 92.82 | 74.66 | 73.03 | 42.50 | 86.96 | 84.04 |
> |          | CIDER  | 55.41 | 85.28 | 87.20 | 77.56 | 91.62 | 52.81 | 89.04 | 67.42 | 80.76 | 72.57 | 84.72 | 66.85 | 81.46 | 70.41 | 81.58 |
> |          | NOODLE |  8.97 | 98.24 | 23.71 | 95.52 | 15.50 | 96.93 | 60.17 | 85.12 | 48.72 | 88.00 | 55.78 | 86.73 | **35.47** | **91.76** | **89.09** |
> | **50%** | SSD+    | 22.23 | 94.66 | 31.14 | 92.26 | 22.30 | 94.59 | 59.41 | 82.33 | 26.91 | 92.99 | 75.70 | 72.32 | 39.62 | 88.19 | 81.02 |
> |          | CIDER  | 93.23 | 50.19 | 96.39 | 54.93 | 98.25 | 54.32 | 93.67 | 52.30 | 95.12 | 52.09 | 93.35 | 52.90 | 95.00 | 52.79 | 77.23 |
> |          | NOODLE | 36.01 | 93.92 | 33.63 | 93.57 | 15.98 | 96.73 | 32.32 | 93.79 | 42.54 | 90.64 | 60.90 | 85.34 | **36.90** | **92.33** | **85.57** |

---

### Official Review · Reviewer_8YkH · 2025-10-31

**Soundness:** 3
**Presentation:** 3
**Contribution:** 2
**Rating:** 6
**Confidence:** 4

**Summary:**

The paper addresses the problem of out-of-distribution (OOD) detection under the presence of label noise. It introduces a framework named NOODLE, which integrates loss correction techniques from noisy-label learning with a low-rank plus sparse decomposition module for feature-level denoising. The method also defines an “OOD-ness” score to measure the deviation of samples from the in-distribution subspace and employs this score for robust distance-based OOD detection. Experiments are conducted on both synthetic and real-world noisy datasets, including CIFAR-10N, CIFAR-100N, and Animal-10N, covering various noise rates and OOD benchmarks.The results include comparisons with standard OOD detection methods and noise-robust learning baselines, as well as ablation studies on different combinations of loss correction and scoring metrics.

**Strengths:**

1. The work is among the first to systematically study OOD detection under noisy-label supervision. The paper clearly articulates why label noise degrades OOD performance.
2. The low-rank + sparse decomposition idea is elegant and conceptually well-motivated: enforcing low-rank structure recovers class-consistent subspaces, while sparse residuals capture noise and anomalies.
3. Clarity and Writing:The paper is well-organized and readable.

**Weaknesses:**

1. Applicability of the Low-Rank Assumption.
The proposed method fundamentally relies on the assumption that class-conditional feature representations are approximately low-rank.
However, this assumption may not hold for fine-grained or multimodal datasets (e.g., ImageNet or text-based tasks).
It is recommended to evaluate the robustness of the method in more complex or diverse scenarios.
2. Lack of Theoretical Analysis.
Although the paper provides an intuitive explanation, it lacks theoretical justification for how the low-rank decomposition improves ID–OOD separability.Incorporating a concise theoretical discussion.
3. Limited Ablation and Comparative Analysis.
While the experimental results convincingly demonstrate the overall effectiveness of NOODLE, the analysis remains somewhat superficial.
The paper mainly reports aggregate performance metrics (FPR95 / AUROC) without deeper ablation to isolate the contribution of each component.
Furthermore, there is no direct comparison with recent robust OOD methods (e.g., ReAct, CSI,CIDER,) under label noise, which would more clearly establish NOODLE’s advantage in this specific setting.
Finally, the paper does not report training stability or computational cost associated with the low-rank decomposition, which would be valuable for assessing practicality.

**Questions:**

1. How stable is the low-rank decomposition (PI-based) during training? Is it recomputed per epoch or per batch? What is its computational overhead relative to baseline methods?
2. Have you tried applying NOODLE on large-scale noisy datasets (e.g., WebVision, Clothing1M)?
3. Could the OOD-ness score be used as an auxiliary regularizer or confidence measure during training rather than only for post-hoc filtering?
4. What happens under asymmetric or instance-dependent noise models? Would the framework generalize naturally?

---

> ### Author Response · Authors · 2025-11-26
> **Response to Reviewer 8YkH**
>
> We would like to thank the reviewer for your reviewing efforts and insightful comments. Please see our detailed response as follows:
>
>
>
> **[Applicability of low-rank assumption and evaluation on large-scale datasets]** We appreciate the reviewer’s comments regarding the low-dimensional assumptions on feature representations for more complex datasets such as ImageNet. We would like to point out that there is strong empirical evidence suggesting that ImageNet representations at the penultimate layers exhibit low-dimensional structure. For example, the work in [1] shows that using only about 5% of the principal components in the penultimate feature space is enough to reach the same level of classification accuracy as using the full ambient dimensions on the ImageNet-1K dataset with a ResNet-50 architecture. To support this observation in the context of label-noise-robust OOD detection, we expand our evaluation to settings involving large-scale datasets, as suggested by the reviewer. Specifically, we evaluate the **NOODLE module on OOD detection tasks under few-shot learning using pre-trained CLIP with ViT-B/16 as the visual encoder and ImageNet-1K as the in-distribution (ID) dataset**. The OOD datasets include iNaturalist, SUN, Places, and Texture. As a baseline, we use CoOp [2], which applies a cross-entropy loss where the predicted probabilities are derived from cosine similarities between the image embeddings and text embeddings:
>
> $\Pr(y = k \mid \mathbf{x}) =$
> $\exp\left( \text{sim}(\mathbf{f}, \mathbf{g}_{k'}) / \tau \right)$
> $/$
>
> $\hspace{3cm} \sum_{k'=1}^{K} \exp\left( \text{sim}(\mathbf{f}, \mathbf{g}_{k'}) / \tau \right)$
>
>
> Here, $\mathbf{f}$ is the visual embedding of the image $\mathbf{x}$, and $\mathbf{g}_k$ is the textual embedding corresponding to class $k$.
>
> We integrate the NOODLE module using the same low-rank plus sparse decomposition on image features as in our other experiments. Both CoOp and NOODLE are fine-tuned under the 16-shot setting. The results show that NOODLE brings meaningful improvements in these large-scale scenarios as well.
>
>
>
>
> ####  Table 1: OOD detection performance on ImageNet-1K under different symmetric noise levels. CLIP with ViT-B/16 as the visual encoder
>
> | Symmetric Noise | Method | iNaturalist FPR ↓ | iNaturalist AUROC ↑ | SUN FPR ↓ | SUN AUROC ↑ | Places365 FPR ↓ | Places365 AUROC ↑ | Texture FPR ↓ | Texture AUROC ↑ | **Avg FPR ↓** | **Avg AUROC ↑** |
> |:---------------:|:--------|:-----------------:|:-------------------:|:----------:|:-------------:|:----------------:|:------------------:|:-------------:|:----------------:|:--------------:|:----------------:|
> | **30%** | Co-Op  | 20.66 | 95.83 | 27.88 | 92.84 | 36.64 | 89.94 | 52.84 | 84.67 | 34.51 | 90.82 |
> |          | NOODLE | 20.89 | 95.72 | 26.81 | 93.22 | 36.48 | 89.85 | 50.39 | 85.26 | **33.64** | **91.01** |
> | **50%** | Co-Op  | 20.60 | 95.67 | 29.00 | 92.64 | 37.25 | 89.72 | 54.29 | 83.07 | 35.29 | 90.78 |
> |          | NOODLE | 18.30 | 96.08 | 26.60 | 93.61 | 35.82 | 90.48 | 46.52 | 85.71 | **33.81** | **91.47** |
> | **75%** | Co-Op  | 20.43 | 96.19 | 33.89 | 92.17 | 40.16 | 89.77 | 52.32 | 85.27 | 36.70 | 90.85 |
> |          | NOODLE | 20.78 | 95.89 | 28.13 | 92.88 | 38.63 | 90.87 | 52.19 | 85.74 | **34.93** | **91.35** |
>
> [1] Feng, Ruili, et al. "Rank diminishing in deep neural networks." Advances in Neural Information Processing Systems 35 (2022): 33054-33065.
> [2] Zhou, Kaiyang, et al. "Learning to prompt for vision-language models." International Journal of Computer Vision 130.9 (2022): 2337-2348.
>
>
> **[Regarding detailed results on main tables]** Thank you for this comment. **In our ablation study figures in Fig. 4 and Fig. 5, we provide a detailed breakdown of each component of the NOODLE framework**. Specifically, the figures separately present the contributions from with and without label-noise robust approach and with and without low-rank sparse decomposition. We also choose different label noise-robust approaches (CM/SCE) as well as different OOD detection scores (KNN/Mahalanobis).
>
> In the revised manuscript, we will make these discussions clearer when discussing these figures as well present the tables in the appendix separately. We will add pointers as well in the main paper.

---

> ### Author Response · Authors · 2025-11-26
> **Response to Reviewer 8YkH**
>
> **[Baselines such as ReAct, CSI,CIDER]** We would like to also point out that we have already included ReAct and CIDER in our original submission in Tables 1 and 2. We will add more explicit pointers to note that we have added these recent OOD methods. For your reference, we are presenting the tables here.
>
>
> ### Table 2: OOD detection performance (FPR95↓ / AUROC↑) on CIFAR-10 with symmetric noise using a ResNet-18 encoder.
>
> | Sym Noise | Method | SVHN FPR | SVHN AUROC | Fashion FPR | Fashion AUROC | LSUN FPR | LSUN AUROC | iSUN FPR | iSUN AUROC | Texture FPR | Texture AUROC | Places365 FPR | Places365 AUROC | Avg FPR | Avg AUROC | ID Acc |
> |:---------:|:--------|:---------:|:-----------:|:-------------:|:---------------:|:----------:|:------------:|:----------:|:-------------:|:--------------:|:----------------:|:----------------:|:------------------:|:--------:|:-----------:|:-----:|
> | **30%** | ReAct | 83.80 | 79.63 | 67.00 | 88.33 | 50.74 | 89.60 | 77.29 | 77.52 | 87.64 | 72.66 | 86.34 | 64.27 | 75.47 | 78.67 | 86.75 |
> |          | CIDER | 55.41 | 85.28 | 87.20 | 77.56 | 91.62 | 52.81 | 89.04 | 67.42 | 80.76 | 72.57 | 84.72 | 66.85 | 81.46 | 70.41 | 81.58 |
> |          | NOODLE | 8.97 | 98.24 | 23.71 | 95.52 | 15.50 | 96.93 | 60.17 | 85.12 | 48.72 | 88.00 | 55.78 | 86.73 | **35.47** | **91.76** | **89.09** |
> | **50%** | ReAct | 79.18 | 77.62 | 72.85 | 85.98 | 42.96 | 92.02 | 95.69 | 58.85 | 91.88 | 52.46 | 91.44 | 59.97 | 79.00 | 71.15 | 82.96 |
> |          | CIDER | 93.23 | 50.19 | 96.39 | 54.93 | 98.25 | 54.32 | 93.67 | 52.30 | 95.12 | 52.09 | 93.35 | 52.90 | 95.00 | 52.79 | 77.23 |
> |          | NOODLE | 36.01 | 93.92 | 33.63 | 93.57 | 15.98 | 96.73 | 32.32 | 93.79 | 42.54 | 90.64 | 60.90 | 85.34 | **36.90** | **92.33** | **85.57** |
>
>
>
>  In addition, we have also added 3 more baselines(FDBD[3], Rankfeat-1[4], NECO[5]) in our study. Please see the  table below.
>
>
> ###  Table 3: OOD detection performance (FPR95↓ / AUROC↑) on CIFAR-10 with real noisy labels using a DenseNet-100 encoder.
>
> | Noise Level | Method      | SVHN FPR | SVHN AUROC | FashionMNIST FPR | FashionMNIST AUROC | LSUN FPR | LSUN AUROC | iSUN FPR | iSUN AUROC | Texture FPR | Texture AUROC | Places365 FPR | Places365 AUROC | Avg FPR | Avg AUROC |
> |:-----------:|:------------|:--------:|:-----------:|:-----------------:|:-------------------:|:--------:|:-----------:|:--------:|:------------:|:------------:|:---------------:|:---------------:|:----------------:|:--------:|:-----------:|
> | **Worst**   | FDBD        | 48.40 | 91.34 | 70.39 | 75.34 | 60.75 | 81.90 | 69.90 | 73.77 | 74.72 | 72.60 | 95.00 | 38.63 | 69.86 | 72.26 |
> |             | RankFeat  | 60.41 | 85.53 | 61.81 | 84.44 | 25.66 | 93.74 | 65.47 | 76.22 | 68.28 | 65.89 | 96.40 | 36.73 | 63.00 | 73.76 |
> |             | NECO        | 78.06 | 60.65 | 70.85 | 70.22 | 58.76 | 75.76 | 83.83 | 62.01 | 74.05 | 71.28 | 74.15 | 73.12 | 73.28 | 68.84 |
> |             | **NOODLE**  | **3.17** | **99.47** | **21.57** | **96.21** | **7.57** | **98.03** | **28.09** | **97.28** | **30.52** | **92.53** | **61.75** | **84.71** | **25.45** | **94.71** |
> | **Agree**   | FDBD        | 71.83 | 83.66 | 46.31 | 87.52 | 47.19 | 89.26 | 79.11 | 74.58 | 78.30 | 69.04 | 88.81 | 61.12 | 68.59 | 77.53 |
> |             | RankFeat  | 24.32 | 94.38 | 48.27 | 91.09 | 14.21 | 97.55 | 70.29 | 77.94 | 61.72 | 74.77 | 92.18 | 57.91 | 51.83 | 82.27 |
> |             | NECO        | 81.73 | 66.91 | 66.42 | 72.13 | 65.12 | 67.84 | 46.48 | 82.13 | 76.79 | 67.93 | 61.95 | 79.76 | 66.42 | 72.79 |
> |             | **NOODLE**  | **1.34** | **99.75** | **17.07** | **97.21** | **2.26** | **98.95** | **8.42** | **98.21** | **17.15** | **96.21** | **52.88** | **88.41** | **16.52** | **96.46** |
> | **Rand1**   | FDBD        | 48.40 | 91.34 | 70.39 | 75.34 | 60.75 | 81.90 | 69.90 | 73.77 | 74.72 | 72.60 | 95.00 | 38.63 | 69.86 | 72.26 |
> |             | RankFeat  | 40.54 | 89.96 | 78.08 | 74.02 | 30.94 | 91.98 | 93.35 | 48.12 | 69.47 | 65.98 | 97.83 | 32.54 | 68.37 | 67.10 |
> |             | NECO        | 87.57 | 55.73 | 50.18 | 82.85 | 53.82 | 80.93 | 68.28 | 78.11 | 70.27 | 77.44 | 64.11 | 81.02 | 65.70 | 76.01 |
> |             | **NOODLE**  | **3.21** | **99.47** | **15.65** | **97.21** | **5.23** | **98.93** | **5.03** | **98.85** | **12.79** | **97.69** | **58.03** | **86.35** | **16.66** | **96.42** |
>
> [3] Liu, L. and Qin, Y. Fast decision boundary based out-of-distribution detector. ICML, 2024.
> [4] Yue Song, Nicu Sebe, and Wei Wang. Rankfeat: Rank-1 feature removal for out-of-distribution detection. Advances in Neural Information Processing Systems, 35:17885–17898, 2022.
> [5] Ammar, M. B., Belkhir, N., Popescu, S., Manzanera, A., and Franchi, G. NECO: NEural collapse based out-of-distribution detection. In The Twelfth International Conference on Learning Representations, 2024.

---

> ### Author Response · Authors · 2025-11-26
> **Response to Reviewer 8YkH**
>
> **[Computational cost]** Thank you for this comment and allowing us to clarify. The added regularization in the NOODLE approach is implemented via a power-iteration (PI)–based low-rank approximation of the batch-wise feature matrix $\boldsymbol{H}$ (see the detailed discussion in Section 3 “Proposed Approach” and Algorithm 1 and 2 in the appendix). Since the PI method uses simple matrix–vector multiplications instead of a full SVD, the computational complexity is $\mathcal{O}\big(T(DBK) + DK^2\big),$
> where $B$ is the batch size, $K$ is the number of classes (or the low-rank parameter), $D$ is the feature embedding dimension, and $T$ is the number of PI iterations (set as 3 in our experiments). **This overhead is moderate and significantly cheaper than exact nuclear-norm optimization for low-rank decompositions**. In experiments, this adds only 5–8 ms per batch, and **the overhead is small compared to the forward and backward passes. We also observe that the overall training time of NOODLE is comparable to GCE/SCE and approximately 4–5 times faster than DivideMix**.
>
>
>
> ### Table 4: Training Time Comparison (100 Epochs + Per-Epoch Time)
>
> | Baseline     | Total Time (100 epochs) | Per-Epoch Time |
> |--------------|--------------------------|-----------------|
> | **GCE**       | 1 hr 42 min              | 1.02 min/epoch  |
> | **SCE**       | 55 min                   | 0.55 min/epoch  |
> | **DivideMix** | 5 hr 30 min              | 3.30 min/epoch  |
> | **Co-teaching** | 2 hr 30 min            | 1.50 min/epoch  |
> | **NOODLE**    | 1 hr 30 min              | 0.90 min/epoch  |
>
>
>
>
>
> **[OOD-ness score]** Thank you for raising this point. Indeed, in our current formulation, the OOD-ness score is **already used as an auxiliary regularizer during training, rather than only as a post-hoc filtering mechanism**. Specifically, we decompose the feature matrix $\boldsymbol{H}$ into an ID component $\boldsymbol{H_{ID}}$
>  and a residual component $\boldsymbol{H_{OOD}}$ , and introduce the column-sparse regularizer as described in Eq. (4) of the main paper. This regularizer explicitly encourages small OOD-ness scores during training, thereby suppressing OOD-like components in the learned representations of the noisy ID dataset. During inference, we additionally use these scores in a post-hoc manner to construct the reference set for the OOD detector, as in Eq. (5). We hope this clarification helps, and we will revise the discussion in the proposed method section to better emphasize this point.
>
>
> **[NOODLE performance under different noise types]** Thank you for your question on different noise scenarios. By design, NOODLE is applicable to different noise types including asymmetric and instance-dependent noise. Irrespective of the noise type, when the feature distortion due to label noise moves away from low-dimensional ID clusters, NOODLE enhances robustness in OOD detection. This is reflected in the following table, where NOODLE improves the performance of the vanilla KNN detector and confusion matrix--based noise robust method CM under both 30% and 50% asymmetric noise.
>
>
> ###  Table 5: OOD detection performance (FPR95 ↓ / AUROC ↑)
> **CIFAR-10 with assymmetric label noise using a DenseNet-100 encoder**
>
> | Asymmetric Noise | Method | SVHN FPR | SVHN AUROC | FashionMNIST FPR | FashionMNIST AUROC | LSUN FPR | LSUN AUROC | iSUN FPR | iSUN AUROC | Texture FPR | Texture AUROC | Places365 FPR | Places365 AUROC | **Avg FPR** | **Avg AUROC** |
> |:----------------:|:--------|:--------:|:-----------:|:----------------:|:--------------------:|:--------:|:-----------:|:---------:|:------------:|:------------:|:---------------:|:--------------:|:----------------:|:-----------:|:-------------:|
> | **30%** | CM | 8.76 | 98.50 | 14.75 | 97.17 | 7.80 | 98.60 | 7.83 | 98.64 | 19.04 | 96.16 | 50.71 | 89.06 | 18.15 | 96.35 |
> |        | KNN | 17.73 | 97.09 | 22.93 | 96.12 | 12.07 | 97.68 | 20.73 | 96.13 | 25.05 | 95.24 | 58.50 | 85.46 | 26.17 | 94.62 |
> |        | **NOODLE** | 4.57 | 99.20 | 22.27 | 95.30 | 3.75 | 99.20 | 9.89 | 98.08 | 15.85 | 96.89 | 50.81 | 87.51 | **17.86** | **96.03** |
> | **50%** | CM | 9.86 | 98.35 | 18.67 | 96.70 | 7.90 | 98.48 | 26.03 | 95.72 | 27.09 | 94.68 | 55.74 | 86.19 | 24.22 | 95.02 |
> |        | KNN | 21.72 | 96.14 | 25.56 | 95.55 | 21.94 | 96.12 | 38.17 | 93.34 | 32.29 | 93.73 | 55.94 | 87.39 | 32.60 | 93.71 |
> |        | **NOODLE** | 7.81 | 98.21 | 22.41 | 96.04 | 9.24 | 98.28 | 11.16 | 97.78 | 27.75 | 93.79 | 52.27 | 86.78 | **21.77** | **95.15** |

---

### Author Response · Authors · 2025-11-28
**Summary of our response**

Dear Reviewers and AC,

Thank you for your reviewing efforts, careful evaluation, and constructive feedback. We hope our responses addressed your concerns and helped clarify the strengths and contributions of our work, **which is the first to offer a principled solution to label noise challenges in OOD detection**. Please let us know if you have more questions and we are happy to address further.

 Meanwhile, we provide a brief summary of our overall response for your reference. We list the major questions raised in your reviews and the corresponding summary of our response.

1. ***Clarification on the rationale of our NOODLE approach***
   - We clarified **the explanation of Fig. 1 on how label noise affects ID-OOD separability**. We also **provided kNN histogram figures (updated Fig. 1 in the revised manuscript)** to further support this idea.
   - We made clarification regarding the $H_{\text{OOD}}$ score and explained that $H_{\text{OOD}}$ **captures any non-ID deviations irrespective of the nature of the shift**.
   - We explained the **robustness of our proposed OOD detector through "hard" sample selection** and showed detailed ablation study of varying the related hyperparameter $p$


2. ***Expanded hyperparameter study and computation cost comparison***
   - We expanded our ablation study to include a more **comprehensive study of key hyperparameters** $p$ **and** $K$.
   - We presented **computation cost comparison table** in the response and showed **less computational overhead for NOODLE** compared to other baselines


3. ***Additional baselines, datasets, and noise types***
   - We extended our experiments to **large-scale OOD detection by applying NOODLE to few-shot CLIP (ViT-B/16) with ImageNet-1K as ID dataset** and showed that the idea of NOODLE is promising to large-scale, challenging scenarios.
   - We presented **additional baselines such as FDBD, RankFeat, and NECO** for different noisy scenarios and showed NOODLE's competitiveness especially in noisy scenarios
   - We updated CIDER and SSD+ across all cases to ensure fair comparison.
   - We presented evaluation on **different noise settings such as asymmetric noise** where NOODLE improves the performance of the vanilla KNN detector and the confusion-matrix–based noise-robust method CM.
   - We have also used **MSP in the inference stage after training under NOODLE to show its effectiveness with logit-based scoring**.

4. ***Manuscript revision and organization***
   - We **updated Fig. 1 in the paper and extended the figure with $k$NN histograms across noise levels** to further support the motivation behind our method.
   - We **updated the section “Low-rank plus Sparse Decomposition”** and added detailed discussions on CM and SCE approaches in "Loss correction" to improve the readability.
   - We **reorganized the tables of the experimental section** to show a clearer division between OOD baselines and label-noise–robust baselines.
   - We added **Remark 1 in the revised manuscript to explain the computational complexity** of the PI-based approach
   - We added explicit pointers to highlight some key tables in the supplementary section
   - We added **additional hyperparameter study tables (Tables 8-10)** in the supplementary section


Sincerely,
Authors

---

### Meta-Review · Area_Chair_MYwD · 2025-12-05

**Summary:**

The reviewers agree that the paper studies an important and underexplored problem: OOD detection under noisy labels. They also find the writing generally clear and the empirical results strong within the chosen setup. **However, several substantial concerns remain.** These include the limited experimental scope (mostly CIFAR-10/CIFAR-100 with DenseNet and training-from-scratch), missing or incomplete baselines, unclear scalability of the proposed low-rank regularization, and questions about the correctness or fairness of some comparisons (e.g., CIDER/SSD+ configuration, hyperparameter sensitivity, and the construction of the OOD reference set). **One reviewer also questions the core interpretation of the low-rank + sparse decomposition and how the sparse component jointly models noisy ID samples and OOD-like deviations.** Although the rebuttal adds more experiments and clarifications, some of the central issues remain only partially resolved.

**Reviewer Concerns:**

**Concerns that the rebuttal addressed**

Reviewer 8YkH raised questions about large-scale evaluation, hyperparameter studies, cost considerations, and the interpretation of the auxiliary score. The authors provided clarifications and additional results, and these points are largely resolved from the reviewer’s perspective.

Reviewer udQU asked about fairness in baseline configurations, the use of MSP-based evaluation, the need for clean-label performance, and the lack of larger model backbones. The authors corrected misconfigurations, added ablations, and supplied several new results. Some concerns are alleviated, although the broader questions on scalability remain.

Reviewer kwRU had several comments on clarity, organization, decomposition motivation, and inclusion of additional baselines. The revised version addresses these comments in a satisfactory way, and this reviewer’s concerns appear to be fully resolved.

**Concerns that remain open**

The experimental scope is still limited. The added large-scale results are only preliminary and do not provide a complete picture of performance in realistic high-dimensional settings.

The scalability of the low-rank formulation is still uncertain. The rebuttal explains the implementation cost in a conceptual manner, but the underlying feasibility concerns raised by Reviewer v1hQ are not fully resolved.

The conceptual foundation of the low-rank plus sparse decomposition remains unclear. The rebuttal does not fully address why noisy in-distribution deviations and true out-of-distribution behavior should be captured by the same sparse component.

Hyperparameter sensitivity remains visible in the added tables and continues to be a point of concern.

Several widely used baselines for modern out-of-distribution detection are still missing from the evaluation despite being explicitly mentioned by the reviewers.

These unresolved issues, especially the conceptual and scalability-related ones, affect the overall confidence in the proposed method.

**Reviewer Scores:**

Reviewer 8YkH initially gave a score of 6. Their concerns were largely addressed, and the tone of the review suggests that the score would likely **remain unchanged**.

Reviewer udQU initially gave a score of 4. Since many of their concrete concerns were resolved, the most reasonable expectation is a modest upward adjustment. **A score of 6 is plausible**.

Reviewer v1hQ initially gave a score of 2. This reviewer expressed fundamental concerns and the rebuttal only partially addressed them. Their score would likely **remain unchanged**.

Reviewer kwRU initially gave a score of 8. All their concerns were addressed. Their score would very likely **remain unchanged**.

---

### Decision · Program_Chairs · 2026-01-26

Reject